# WINFORMER: TRANSCENDING PAIRWISE SIMILARITY FOR TIME-SERIES GENERATION

## ABSTRACT

Time-series generation plays a critical role in data imputation, feature augmentation, domain adaptation, and foundation modeling. However, the cross-domain generation remains a persistent challenge, as existing methods model time-series interactions either at the granularity of individual points or fragmented segments. This limits their ability to capture and adapt to complex periodic patterns inherent in diverse domains. Specifically, point-wise attention struggles with long-range dependencies, while standard patch-based approaches may break important cyclical structures. To address this, we introduce Winformer, a novel diffusion model framework built on a window-wise Transformer. We shift the fundamental processing unit in the attention mechanism from pairwise points similarity to continuous windows comparison of the entire horizon. By operating on semantically richer window representations, the proposed approach effectively learns and transfers complex periodic patterns across domains. Extensive experiments on 12 real-world datasets demonstrate Winformer's effectiveness, achieving an average performance gain of 10.67% over SOTA baselines.

## 1 INTRODUCTION

The generation of high-quality time-series becomes a fundamental step in data imputation (Alcaraz & Strodthoff, 2023; Tashiro et al., 2021), feature augmentation (Senane et al., 2024; Yuan & Qiao, 2024), domain adaptation (Zhang et al., 2025; Huang et al., 2025) and foundation modeling (Ma et al., 2024; Cao et al., 2025). The difficulty arises from the complex distribution of trends, period and noise that evolve along with time. As some researchers (Huang et al., 2025) have pointed out, the distribution becomes particularly complex during cross-domain generation.

Following the vanilla Transformer design (Vaswani et al., 2017; Zhou et al., 2021), the traditional methods (Yuan & Qiao, 2024; Piao et al., 2024; Zhou et al., 2022) model the exact interactions between individual time-series through a point-wise perspective, as illustrated in Fig. 1(a). The long-range and high-order interactions are captured with frequency components (Zhou et al., 2022) and stacking of attention layers (Wu et al., 2021). Recently, the time-series patching (Nie et al., 2023; Peebles & Xie, 2023) is introduced to obtain middle-level interactions. However, the equidistant patching directly breaks evolving trends and periodic patterns, as in Fig. 1(b), where the RevIN (Kim et al., 2022) technique is necessarily applied to restore the distribution shifting. The point-wise and patch-wise time-series partitioning originates from the pairwise similarity design of the attention mechanism, and we find that a more adaptive architecture works better for coupling trending and periodic patterns. To better demonstrate the effectiveness, we focus on the more challenging cross-domain generation tasks with domain specific time-series interactions.

To address this issue, we extend the fundamental principle of attention mechanism, i.e. pairwise similarity, to windows comparison. As shown in Fig. 1, the slicing window of time-series is set as 1 in point-wise modeling and the attention is calculated on pairwise points, while the patch-wise window is set as 3 and the attention is forced on pairwise lattices. Unlike the window slicing operation used in traditional data processing, our approach provides a new way to adopt sliding window-based comparison during attention calculation, thereby enhancing adaptive capability. We propose the Ample attention to compare the Fourier-deduced similarity between time-series windows in Fig. 1(c), it enables a learnable neural operator for domain adaptability. Our contributions are:

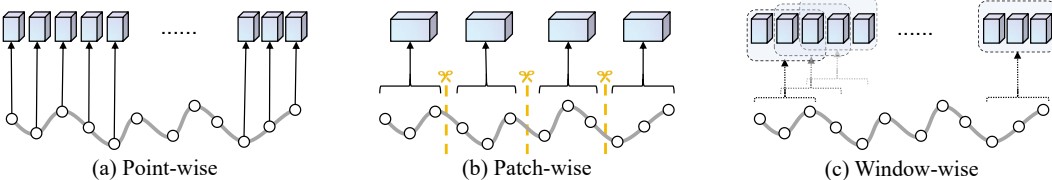

Figure 1: The perspective of time-series modeling. (a) Point-to-point modeling on exact interactions (Slicing window, window size=1). (b) Patch-to-patch modeling on middle-level interactions (Slicing window, window size=3). (c) We propose window-to-window modeling on complex interactions (Sliding window, window size=3).

- We propose the Ample attention, which transcends the calculation of statistic pairwise similarity in vanilla Transformer. It is deduced from the comparison of time-frequency and is calculated through learnable convolution operator on original attention.
- We design the Winformer, a new transformer-based denoising architecture, leveraging the window-wise attention to enhance the learning and generation of domain specific time-series interactions.
- Empirical experiments demonstrates 10.67% performance improvement on 12 real-world datasets against SOTA baselines.

## 2 PRELIMINARY

With the rolling forecasting setting with a fixed horizon, we define the $t$-th sequence inputs of domain $j$ as $\mathcal{X}^{(j,t)} = \{\mathbf{x}_1^{(j,t)}, \ldots, \mathbf{x}_{L_x}^{(j,t)} \mid \mathbf{x}_i^{(j,t)} \in \mathbb{R}^D\}$, where $L_x$ stands for the horizon length. We mix the $M$ datasets from different domains together to get the unified dataset $\mathcal{X} = \cup_{i=1}^M \mathcal{X}_i$ considering the cross-domain time-series generation settings, and the overall target is to learn a parameterized model $\theta$ with $p_\theta(\mathbf{x}_1, \mathbf{x}_2, ..., \mathbf{x}_T | i)$.

### 2.1 VANILLA SELF-ATTENTION

The self-attention mechanism (Vaswani et al., 2017) yields successful pairwise alignment ability in sequence modeling. It is calculated on three transformed inputs from $\mathbf{X} \in \mathbb{R}^{L_x \times D}$, i.e, query, key and value, which is defined as the scaled dot-product as:

$$\mathcal{A}(\mathbf{Q}, \mathbf{K}, \mathbf{V}) = \text{Softmax}(\frac{\mathbf{Q}\mathbf{K}^\top}{\sqrt{d}})\mathbf{V} \qquad , \tag{1}$$

where we have $\mathbf{Q} = \mathbf{X}\mathbf{W}_Q^\top$, $\mathbf{K} = \mathbf{X}\mathbf{W}_K^\top$, $\mathbf{V} = \mathbf{X}\mathbf{W}_V^\top$ with trivial projections $\mathbf{Q} \in \mathbb{R}^{L_Q \times d}$, $\mathbf{K} \in \mathbb{R}^{L_K \times d}$, $\mathbf{V} \in \mathbb{R}^{L_V \times d}$ and the efficient scaled norm of the input dimension $d$.

### 2.2 DENOISING DIFFUSION PROBABILISTIC MODELS

The diffusion probabilistic generation methods are based on the assumption of Markov chain. Specifically, let the input matrix $\mathbf{X}_{(0)} \in \mathbb{R}^{L_x \times D} \sim q(\mathbf{X})$ be the real data. At the $n$ step of the diffusion process, we obtain $\mathbf{X}_{(n)}$ from $\mathbf{X}_{(n-1)}$ by adding Gaussian noise with the transition kernel defined as $q(\mathbf{X}_{(n)} | \mathbf{X}_{(n-1)}) = \mathcal{N}(\mathbf{X}_{(n)}; \sqrt{1-\beta_n}\mathbf{X}_{(n)}, \beta_n \mathbf{I})$, where $\beta_N \in (0,1)$. By recurring the $n$ steps of the Markov chain, we can derive:

$$q(\mathbf{X}_{(n)} | \mathbf{X}_{(0)}) = \mathcal{N}\left(\mathbf{X}_{(n)}; \sqrt{\bar{\alpha}_n}\mathbf{X}_{(n)}, (1-\bar{\alpha}_n)\mathbf{I}\right) \qquad , \tag{2}$$

where $\alpha_n = 1 - \beta_n$ and $\bar{\alpha}_n = \prod_{i=1}^n \alpha_i$. In reversing, a learning-based model reconstructs as $p_\theta(\mathbf{X}_{(n-1)} | \mathbf{X}_{(n)}) = \mathcal{N}\left(\mathbf{X}_{(n-1)}; \mu_\theta(\mathbf{X}_{(n)}, n), \Sigma_\theta(\mathbf{X}_{(n)}, n)\right)$ and we estimate $\mu$:

$$\mu_\theta(\mathbf{X}_{(n)}, n) = \frac{1}{\sqrt{\alpha_n}}\left(\mathbf{X}_{(n)} - \frac{\beta_n}{\sqrt{1-\bar{\alpha}_n}}\epsilon_\theta(\mathbf{X}_{(n)}, n)\right) \qquad . \tag{3}$$

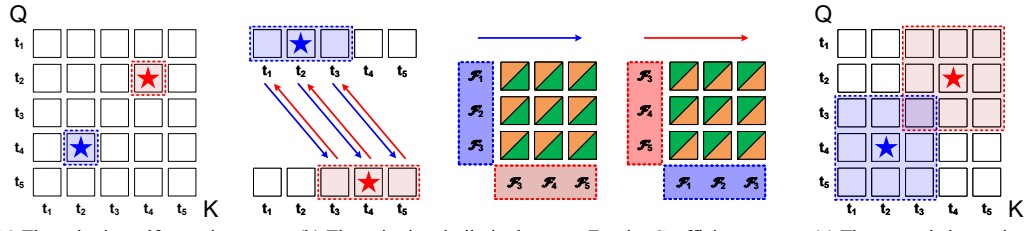

(a) The pairwise self-attention.  (b) The pairwise similarity between Fourier Coefficients.  (c) The expanded attention.

Figure 2: The process of attention expansion. (a) The visualization of pairwise feature map (dot-product) of vanilla self-attention. There are five tokens ($t_1, t_2, t_3, t_4, t_5$) in both query and keys. And the blue star ($Q_{t_4}$, $K_{t_2}$) and red star ($Q_{t_2}$, $K_{t_4}$) form a duality. (b) The new similarity is defined between the windowed Fourier coefficients. We set the window size as 3 and the matrix is the dot-product of corresponding coefficients, where the orange triangle stands for real parts and the green triangle for imaginary parts. The blue arrow denotes the grouped similarity from ($t_1, t_2, t_3$) to ($t_3, t_4, t_5$) like the blue star in figure a. (c) We design the expanded attention to crop the scores from original feature map.

## 3 METHODS

In this section, we first introduce the derivation of the proposed Ample attention and its calculation. Then, we describe the overall Winformer architecture. Some theorems mentioned in this section are not further elaborated on due to space limitation, and we provide theory backgrounds in Appendix G.

### 3.1 THE WINDOW-WISE MODELING

The vanilla self-attention mechanism is built upon the point-to-point dot-product similarity. It achieves significant alignment ability over the tokenized inputs of language (Devlin et al., 2019), where the text embedding Word2Vec (Mikolov et al., 2013) and Jina (Günther et al., 2023) ensure the unified linguistic space. For time-series generation, the Fourier transform decomposes the sequences into constituent frequencies and defines similarity between the frequency coefficients to make it possible to align the series. Building on this, we propose transforming time-series data into the Fourier basis and calculating similarity over complex planes.

#### 3.1.1 WINDOW TO WINDOW ALIGNMENT

The time-series data exits heavily warping problem (Berndt & Clifford, 1994; Salvador & Chan, 2007). This problem arises when comparing or analyzing different time-series sequences, where similar or identical events do not occur at precisely the same time steps across these series. In other words, the event timestamps lead to misalignment, which can be caused by various factors, including differences in sampling rates, system delays, manual operation delays, or inconsistencies in natural changing rates. To alleviate the time warping, we replace the point-to-point calculation with the window-to-window one.

Recalling that we have an input matrix $\mathbf{X}$, the $i$-th time step is represented by a vector $\mathbf{x}_i \in \mathbb{R}^{1 \times D}$. Assume that we perform a window-wise Fourier transformation, over these finite-length signals, and the window size is set as $p$. Since the time-series become discrete-time inputs, we use the discrete Fourier transform (DFT) $\mathcal{F}_D$ instead. We apply the DFT operator to each feature dimension independently, which follows the same setting (Alaa et al., 2021). We select the $q$-th dimension of projected inputs $\mathbf{Q}$ as $\mathbf{Q}^q = \{\mathbf{x}_1^q, \ldots, \mathbf{x}_{L_x}^q\}$. Through the temporal zero padding on the beginning and ending, we have the window-wise attention inputs $\mathbf{Q}^q$ and $\mathbf{K}^q$ respectively:

$$[\mathbf{Q}]_i^{(q)} = \{\mathbf{Q}_j^q | i \times s \leq j < (i+1) \times s\} \qquad , \tag{4}$$

and

$$[\mathbf{K}]_i^{(q)} = \{\mathbf{K}_j^q | i \times s \leq j < (i+1) \times s\} \qquad , \tag{5}$$

where the $[\mathbf{Q}]_i^{(q)} \in \mathbb{R}^{s \times 1}$, $[\mathbf{K}]_i^{(q)} \in \mathbb{R}^{s \times 1}$ and $s$ stands for the stride. For the $t$-th time step, the DFT transforms the real-valued inputs into the complex-valued ones as $\mathcal{F}_D\{[\mathbf{Q}]_t^{(q)}\}$ and $\mathcal{F}_D\{[\mathbf{K}]_t^{(q)}\}$.

Likewise the self-attention mechanism, we can perform the dot-product between the selected time step $t_1$ and $t_2$ in the frequency domain, whose scores calculate the similarity between different spectral components (Zhou et al., 2022; Piao et al., 2024; Kong et al., 2023) as:

$$\widetilde{\mathbf{F}}^{(p,q)}_{(t_1,t_2)} = \mathcal{F}_D\{[\mathbf{Q}]^{(q)}_{t_1}\} \cdot \mathcal{F}_D\{[\mathbf{K}]^{(q)}_{t_2}\} \qquad . \tag{6}$$

This score is a complex matrix containing the real component $\mathbf{Re}(\widetilde{\mathbf{F}})$ and imaginary component $\mathbf{Im}(\widetilde{\mathbf{F}})$, a two-channel matrix with shape $2 \times s \times 1$. The former component measures the magnitude difference between the spectral coefficients, while the latter one is about the phase difference.

Then, we can concatenate all the inputs' DFT result as $\widetilde{\mathbf{F}}^{(p)}_{(t_1,t_2)} = [\widetilde{\mathbf{F}}^{(p,1)}_{(t_1,t_2)}, \ldots, \widetilde{\mathbf{F}}^{(p,d)}_{(t_1,t_2)}]$ with a shape of $2 \times s \times d$. Likewise the self-attention mechanism, we take the magnitude difference of the spectral coefficients at $t_1$ and $t_2$ as an example, which could be normalized by $\mathrm{Softmax}(\cdot)$ to reorganize $\mathbf{V}$. Then we can define the real-valued similarity within the window-to-window comparison as:

$$\mathbf{S}^{\mathbf{Re}}_{(t_1,t_2)} = \mathbf{Re}(\widetilde{\mathbf{F}}^{(p)}_{(t_1,t_2)})\mathbf{1} \qquad , \tag{7}$$

where $\mathbf{1} \in \mathbb{R}^{d \times 1}$ denotes the all-one vector and it sums the real coefficient of all channels (also applies to imaginary).

As illustrated in Fig. 2, the original self-attention leverages the pairwise similarity to measure the relationship between different inputs, e.g., the blue star represents the attention feature map for $\mathbf{Q}_{t_4}$ and $\mathbf{K}_{t_2}$ pairs and red star is the dual score. If we perform the windowed DFT and measure the "group" similarity between $(t_1, t_2, t_3)$ and $(t_3, t_4, t_5)$ through the real-valued similarity defined in Eq.(7) and its imaginary ones, we can acquire the blue arrows composed with real parts (the orange triangle) and imaginary parts (the green triangle). Based on the following derivation, we will demonstrate that the pairwise attention could be expanded to crop scores, where the score (blue star) stands for window-to-window comparison.

### 3.1.2 The Convolutional Calculation

The Fourier Transform, a mathematical operator (Duhamel & Vetterli, 1990), can be calculated in Eq.(6), but it requires two transformations, namely product in frequency domain and composition of real part and imaginary part. Actually, our ultimate goal is to calculate a new similarity score in window-wise perspective that is numerically compatible with other network components. Inspired by the convolution theorem (McGillem & Cooper, 1991) which bridges the connection between Fourier operator and convolution operator, we decompose the calculation of the DFT operator as a linear transformation:

$$\mathcal{F}_D(\mathbf{x}) = \mathbf{Mx} \quad , \text{ where}$$

$$\mathbf{M} = \frac{1}{\sqrt{p}} \begin{bmatrix} 1 & 1 & 1 & \cdots & 1 \\ 1 & \omega & \omega^2 & \cdots & \omega^{p-1} \\ 1 & \omega^2 & \omega^4 & \cdots & \omega^{2(p-1)} \\ \vdots & \vdots & \vdots & \ddots & \vdots \\ 1 & \omega^{p-1} & \omega^{2(p-1)} & \cdots & \omega^{(p-1)^2} \end{bmatrix} \qquad . \tag{8}$$

The coefficient $\omega$ is $e^{-2\pi j/p}$. Using the Euler's rule and the $i$-th row of transformation $\mathbf{M}$ be formulated as $\psi(i) = [1, \cos(2\pi i/p), \ldots, \cos(2\pi i(p-1)/p)]$ , we can rewrite $\mathbf{M}$ as $[\psi(0), \ldots, \psi(p-1)]^\top/\sqrt{p}$, and the window-wise similarity score is the sum of different groups:

$$\mathbf{S}_{(t_1,t_2)} = \frac{\mathbf{W}^\top}{\sqrt{p}} \left[ \begin{pmatrix} \psi(0) \\ \vdots \\ \mathbf{0} \end{pmatrix} + \cdots + \begin{pmatrix} \mathbf{0} \\ \vdots \\ \psi(p-1) \end{pmatrix} \right] \begin{pmatrix} \varphi_0 \\ \vdots \\ \varphi_{p-1} \end{pmatrix} \qquad , \tag{9}$$

where the $\mathbf{W}$ is the coefficient for decomposed and reorganized $p$ sub-matrix and the $\mathbf{0}$ is the all zero matrix.

We noticed that the pairwise attention scores $\mathbf{S}'_{(t_1,t_2)}$ has been already calculated in Eq.(1). The target window-wise score $\mathbf{S}_{(t_1,t_2)}$ can be acquired by performing convolution whose shape is larger

than or equal to the periodic one in Eq.(9). Let $\varphi = [\varphi_0, \varphi_1, \ldots, \varphi_{p-1}]$, we have:

$$
\begin{aligned}
\mathbf{S}_{(t_1,t_2)} &= \frac{\mathbf{W}^\top}{\sqrt{p}} \sum [\psi(0)\varphi + \cdots + \psi(p-1)\varphi] \\
&= \frac{\mathbf{W}^\top}{\sqrt{p}} \sum [\text{avg}(\mathbf{S}^{'}) + \text{conv}_{\psi(1)}(\mathbf{S}^{'}) + \cdots + \text{conv}_{\psi(p-1)}(\mathbf{S}^{'})]
\end{aligned}
, \tag{10}
$$

where $\text{conv}_{\psi(i)}(\cdot)$ represents the convolution operator with the kernel $\psi(i)$, with the same basis of DFT in Eq.(8). Thus, we can acquire the window-wise score by performing the convolution operator on the original attention score. Specially, for $\psi(0)$, we can use an all-one matrix as the kernel, which equals that we only select the $\text{avg}(\cdot)$ operator.

### 3.1.3 THE AMPLE ATTENTION

Recalling the vanilla Transformer's attention mechanism, we still follows the $\text{Softmax}(\cdot)$ design while replace the pairwise similarity with the window-wise comparison. We applied generalized Parseval's theorem (Hardy & Titchmarsh, 1931) between the pairwise $\mathbf{S}^{'}_{(t_1,t_2)}$ and window-wise $\mathbf{S}_{(t_1,t_2)}$, it reveals a linear connection on the sum of scores. It motivates us to leverage the $\text{Conv2d}(\cdot)$ layer with learnable kernel, we initialized the kernel with a decomposing basis for fast convergence, then the distribution of linear coefficients $\mathbf{W}$ is learned through convolution kernel. In this way, we can defined the Ample Attention as:

$$
\text{Attn} = \text{Softmax}\left(\text{Conv2d}_\psi(\mathbf{Q}\mathbf{K}^\mathrm{T})\mathbf{V}\right) , \tag{11}
$$

where the kernel $\psi$ could be initialized by an exact type of transformation. If we only consider the term of $\psi(0)$ in Eq. 10, we can perform $\text{Avgpool2d}(\cdot)$ to replace the $\text{Conv2d}_\psi(\cdot)$. In practice, we simplify the window-wise similarity to measure the amplitude correlation and omit phase consistency in imagery part. Also, since the Fourier basis involves imaginary calculations, other frequency decompose bases, such as Discrete Cosine Transform (DCT), are also acceptable for kernel initializing. This is because of the learnable ability of $\text{Conv2d}_\psi(\cdot)$ which looses the constraints of kernel initialization. To this end, we have derived the Ample attention.

## 3.2 TRANSFORMER-BASED DENOISING ARCHITECTURE

The predominant Transformer-based model shows its potential integration capability with the diffusion framework in time-series generation (Ge et al., 2025; Cao et al., 2025). To enhance the learning ability of domain specific time-series interactions through the proposed Ample attention, we introduce a transformer-based denoising architecture, namely Winformer.

### 3.2.1 THE DIFFUSION PROCESS

As shown in Fig. 3, we utilize a conditional diffusion process with a forward process adding Gaussian noise gradually to a data sample $\mathbf{X}_{(0)}$. We parameterize Eq.(2) to sample $\mathbf{X}_{(t)} = \sqrt{\bar{\alpha}_t}\mathbf{X}_{(0)} + \sqrt{1 - \bar{\alpha}_t}\epsilon_t$, where $\bar{\alpha}_t$ are the constant hyper-parameters and $\epsilon_t \sim \mathcal{N}(0, \mathbf{I})$. To invert the forward process, a transformer-based denoising model is trained to estimate the $\mu_\theta$ and calculate out the $\epsilon_\theta$ as Eq.(3) to reconstruct $\mathbf{X}_{(n-1)}$ from $\mathbf{X}_{(n)}$.

### 3.2.2 THE HYBRID ENCODER

We apply the transformer-based denoising model to simulate $\mu_\theta$. For the $k$-th diffusion step, the noisy data $\mathbf{X}_{(k)}$ is embedded with position information. Then the embedded data is processed by $L+1$ encoder blocks, including one hybrid encoder and $L$ DiT encoders. The encoder blocks utilize the adaptive layer normalization mechanism (Peebles & Xie, 2023) to fuse conditional features $c$ with noisy data $\mathbf{X}_{(k)}$. Specially, for the hybrid encoder block, we divide the attention heads into two groups, including the vanilla attention heads and the Ample attention heads. The vanilla attention heads preserve pairwise similarity in Eq.(1), and the Ample attention is calculated on corresponding vanilla ones with window-wise alignment. These two kinds of heads are concatenated in head-level.

As the previous derivation along with Eq.(11), we can initialize the kernel of $\text{Conv2d}_\psi(\cdot)$ with an exact frequency decompose basis for window-wise alignment. We choose the discrete cosine

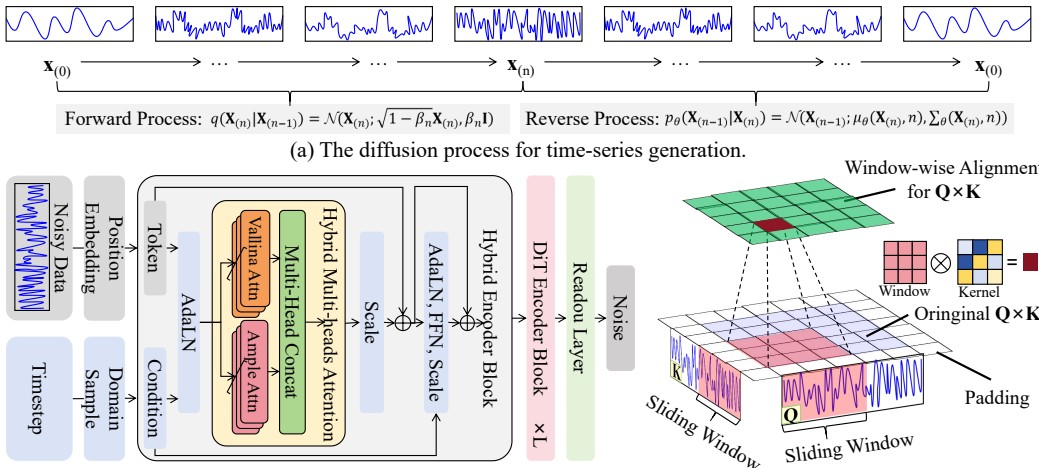

(a) The diffusion process for time-series generation.

(b) Transformer-based denoising model with Ample attention.     (c) Window-wise alignment for Ample attention.

Figure 3: **The overall architecture.** (a) In the forward process, we add Gaussian noise to original time-series data to obtain noisy data. Then, the learning-based denoising model predicts the noise for time-series recovering during the reverse process. (b) The denoising model is based on recent diffusion transformer, which processes the noisy data with several encoder blocks. To enhance the periodic patterns, we reformed the first encoder block with hybrid multi-heads attention, which integrates the Ample attention heads with the vanilla self-attention heads. (c) Different from the vanilla self-attention mechanism, the Ample attention heads conduct window-to-window alignment in similarity score's calculation to capture multiple periodic patterns.

transform (DCT) dbasis, a widely used transformation with real values only, which is depicted as:

$$
\psi(i,j) = \begin{cases} \sqrt{\dfrac{1}{p}} \cos\left[\dfrac{(j+0.5\pi)}{p} i\right] & i = 0 \\[2ex] \sqrt{\dfrac{2}{p}} \cos\left[\dfrac{(j+0.5\pi)}{p} i\right] & i \neq 0 \end{cases}. \tag{12}
$$

### 3.2.3 DOMAIN CONDITIONING

Inspired by TimeDP (Huang et al., 2025), we select series samples to construct domain conditions $c_d$ to acknowledge the model with domain-specific features, and learned prototypes $c_p$ to store basic pattens. Then we combine these conditions with the diffusion step $t$ to obtain the condition $c$ as :

$$
c = \text{Emb}_t(t) + \text{Emb}_{c_p}(c_p) + \text{Emb}_{c_d}(c_d) \qquad , \tag{13}
$$

where $\text{Emb}_t(\cdot)$, $\text{Emb}_{c_p}(\cdot)$ and $\text{Emb}_{c_d}(\cdot)$ represent the embedding methods. The condition instructs the model learning through the adaptive layer normalization process.

## 4 EXPERIMENTS

In this section, we empirically demonstrate Winformer's effectiveness on the twelve real-world datasets and further discuss its generating process through visualization. We also provide more experimental details, including hyper-parameter tests, computational costs, visualization figures and domain discrepancy analysis in Appendix A to Appendix D.

### 4.1 SETUP: CROSS-DOMAIN TIME-SERIES GENERATION

**(a) Datasets.** We conduct experiments on 12 real-world time-series datasets across four domains following TimeDP (Huang et al., 2025), including traffic flows, weather phenomena, industrial logs

Table 1: Results of generation results for sequence length 168. Best results are **bold** and second best results are underlined. Our method outperforms SOTA baselines in most of the datasets and achieve an increase of averagely 10.67% in MMD.

| | | Winformer | TimeDP | Diffusion-TS | TimeGAN | GT-GAN | TimeVAE | TimeVQVAE |
|---|---|---|---|---|---|---|---|---|
| Maximum Mean Discrepancy | Electricity | $\mathbf{0.001}_{\pm 0.002}$ | $\mathbf{0.001}_{\pm 0.001}$ | $0.003_{\pm 0.002}$ | $0.367_{\pm 0.255}$ | $0.254_{\pm 0.166}$ | $0.577_{\pm 0.006}$ | $0.152_{\pm 0.024}$ |
| | Solar | $\mathbf{0.035}_{\pm 0.004}$ | $\underline{0.041}_{\pm 0.011}$ | $0.050_{\pm 0.012}$ | $0.628_{\pm 0.053}$ | $0.578_{\pm 0.039}$ | $0.353_{\pm 0.014}$ | $0.437_{\pm 0.020}$ |
| | Wind | $\underline{0.034}_{\pm 0.014}$ | $\mathbf{0.025}_{\pm 0.017}$ | $0.035_{\pm 0.006}$ | $0.213_{\pm 0.017}$ | $0.170_{\pm 0.040}$ | $0.170_{\pm 0.004}$ | $0.131_{\pm 0.014}$ |
| | Traffic | $\mathbf{0.071}_{\pm 0.005}$ | $0.083_{\pm 0.034}$ | $0.111_{\pm 0.031}$ | $0.567_{\pm 0.057}$ | $0.538_{\pm 0.078}$ | $0.218_{\pm 0.007}$ | $0.213_{\pm 0.016}$ |
| | Taxi | $\mathbf{0.085}_{\pm 0.010}$ | $\underline{0.095}_{\pm 0.023}$ | $0.131_{\pm 0.014}$ | $0.275_{\pm 0.054}$ | $0.319_{\pm 0.032}$ | $0.139_{\pm 0.007}$ | $0.128_{\pm 0.004}$ |
| | Pedestrian | $\mathbf{0.040}_{\pm 0.008}$ | $\underline{0.044}_{\pm 0.020}$ | $0.071_{\pm 0.019}$ | $0.090_{\pm 0.030}$ | $0.112_{\pm 0.019}$ | $0.065_{\pm 0.002}$ | $0.067_{\pm 0.007}$ |
| | Air | $\mathbf{0.011}_{\pm 0.002}$ | $\mathbf{0.011}_{\pm 0.003}$ | $\underline{0.022}_{\pm 0.011}$ | $0.120_{\pm 0.045}$ | $0.211_{\pm 0.041}$ | $0.089_{\pm 0.016}$ | $0.028_{\pm 0.002}$ |
| | Temperature | $0.230_{\pm 0.021}$ | $\mathbf{0.219}_{\pm 0.022}$ | $\underline{0.241}_{\pm 0.049}$ | $0.926_{\pm 0.042}$ | $0.809_{\pm 0.081}$ | $1.002_{\pm 0.014}$ | $0.323_{\pm 0.008}$ |
| | Rain | $\mathbf{0.036}_{\pm 0.016}$ | $0.057_{\pm 0.039}$ | $0.079_{\pm 0.058}$ | $0.329_{\pm 0.285}$ | $0.111_{\pm 0.109}$ | $0.292_{\pm 0.019}$ | $0.074_{\pm 0.007}$ |
| | NN5 | $\mathbf{0.147}_{\pm 0.008}$ | $\underline{0.164}_{\pm 0.010}$ | $0.186_{\pm 0.043}$ | $0.874_{\pm 0.088}$ | $0.632_{\pm 0.074}$ | $0.821_{\pm 0.061}$ | $0.327_{\pm 0.012}$ |
| | Fred-MD | $\mathbf{0.002}_{\pm 0.001}$ | $\mathbf{0.002}_{\pm 0.001}$ | $0.006_{\pm 0.002}$ | $0.043_{\pm 0.021}$ | $0.133_{\pm 0.102}$ | $0.059_{\pm 0.008}$ | $0.008_{\pm 0.002}$ |
| | Exchange | $\mathbf{0.137}_{\pm 0.012}$ | $0.151_{\pm 0.024}$ | $\underline{0.206}_{\pm 0.035}$ | $0.530_{\pm 0.154}$ | $0.475_{\pm 0.116}$ | $0.543_{\pm 0.149}$ | $0.342_{\pm 0.050}$ |
| K-L Divergence | Electricity | $\mathbf{0.008}_{\pm 0.010}$ | $\underline{0.012}_{\pm 0.016}$ | $0.315_{\pm 0.247}$ | $0.488_{\pm 0.175}$ | $0.407_{\pm 0.079}$ | $0.734_{\pm 0.023}$ | $0.280_{\pm 0.051}$ |
| | Solar | $\mathbf{0.013}_{\pm 0.005}$ | $\underline{0.016}_{\pm 0.005}$ | $0.066_{\pm 0.055}$ | $0.612_{\pm 0.447}$ | $0.120_{\pm 0.041}$ | $0.260_{\pm 0.016}$ | $0.865_{\pm 0.108}$ |
| | Wind | $0.202_{\pm 0.044}$ | $\underline{0.152}_{\pm 0.034}$ | $0.548_{\pm 0.372}$ | $1.924_{\pm 1.233}$ | $\mathbf{0.107}_{\pm 0.016}$ | $0.484_{\pm 0.015}$ | $0.483_{\pm 0.066}$ |
| | Traffic | $\underline{0.011}_{\pm 0.002}$ | $\mathbf{0.009}_{\pm 0.003}$ | $0.120_{\pm 0.074}$ | $1.305_{\pm 0.320}$ | $1.409_{\pm 0.251}$ | $0.211_{\pm 0.014}$ | $0.178_{\pm 0.026}$ |
| | Taxi | $\mathbf{0.005}_{\pm 0.003}$ | $\underline{0.011}_{\pm 0.004}$ | $0.075_{\pm 0.034}$ | $0.650_{\pm 0.180}$ | $0.950_{\pm 0.197}$ | $0.110_{\pm 0.020}$ | $0.110_{\pm 0.026}$ |
| | Pedestrian | $\mathbf{0.009}_{\pm 0.004}$ | $\underline{0.014}_{\pm 0.010}$ | $0.133_{\pm 0.069}$ | $0.417_{\pm 0.181}$ | $0.411_{\pm 0.096}$ | $0.065_{\pm 0.005}$ | $0.405_{\pm 0.051}$ |
| | Air | $\mathbf{0.026}_{\pm 0.011}$ | $\underline{0.027}_{\pm 0.016}$ | $0.106_{\pm 0.079}$ | $0.348_{\pm 0.093}$ | $0.578_{\pm 0.049}$ | $0.164_{\pm 0.012}$ | $0.054_{\pm 0.012}$ |
| | Temperature | $\underline{0.176}_{\pm 0.027}$ | $\mathbf{0.171}_{\pm 0.073}$ | $0.342_{\pm 0.131}$ | $8.892_{\pm 2.681}$ | $3.174_{\pm 2.685}$ | $2.183_{\pm 0.110}$ | $0.735_{\pm 0.066}$ |
| | Rain | $\mathbf{0.011}_{\pm 0.003}$ | $\underline{0.013}_{\pm 0.012}$ | $0.061_{\pm 0.072}$ | $0.506_{\pm 0.174}$ | $0.432_{\pm 0.099}$ | $0.160_{\pm 0.022}$ | $0.047_{\pm 0.018}$ |
| | NN5 | $\mathbf{0.045}_{\pm 0.007}$ | $\underline{0.054}_{\pm 0.014}$ | $0.165_{\pm 0.076}$ | $4.928_{\pm 4.112}$ | $1.386_{\pm 0.520}$ | $1.337_{\pm 0.220}$ | $1.063_{\pm 0.274}$ |
| | Fred-MD | $\mathbf{0.201}_{\pm 0.014}$ | $\underline{0.203}_{\pm 0.035}$ | $0.835_{\pm 0.554}$ | $0.512_{\pm 0.290}$ | $0.380_{\pm 0.070}$ | $0.346_{\pm 0.041}$ | $0.831_{\pm 0.077}$ |
| | Exchange | $\mathbf{1.621}_{\pm 0.126}$ | $\underline{1.866}_{\pm 0.132}$ | $2.337_{\pm 0.714}$ | $8.861_{\pm 3.397}$ | $7.201_{\pm 4.380}$ | $10.404_{\pm 1.434}$ | $5.052_{\pm 1.385}$ |
| | Count | 19 | 7 | 0 | 0 | 1 | 0 | 0 |

and financial records. All these datasets are reformatted into non-overlapping uni-variance sequence slices with the length of 168. For cross-domain generation, all datasets are mixed during training.

**(b) Baselines.** We compare our model with 6 representative SOTA methods for cross-domain time-series generation. These methods include GAN-base methods, such as TimeGAN (Yoon et al., 2019) and GT-GAN (Jeon et al., 2022), VAE-based methods, such as TimeVAE (Desai et al., 2021) and TimeVQVAE (Lee et al., 2023), and diffusion-based method, such as the newly released TimeDP (Huang et al., 2025) and Diffussion-TS (Yuan & Qiao, 2024). To ensure a fair comparison, we adopt the related results reported by TimeDP (Huang et al., 2025).

**(c) Metrics.** We select two metrics to evaluate the performance of generation by measuring the similarity between the distributions of the real and generated time-series. **Maximum Mean Discrepancy (MMD)** compares the discrepancy between two series after mapping into a high-dimension feature space with a kernel function. **Kullback-Leibler Divergence (K-L)** measures the divergence between two probability distributions. The metric computation is stated in Appendix E.1.

**(d) Implementation.** As analyzed in Section 3, the stride length of window-alignment is a hyper-parameter, and we choose 25 in our main experiments, which is suitable due to it is larger than common periodic cycles in real-world datasets, such as 4, 6, 12 and 24. We further explored the effectiveness with different stride length and kernel type of window-alignment in Appendix A. The number of DiT encoder layers is set to 6, and the hidden size is 512. The learning rate is set to $5 \times 10^{-5}$ with $1,000$ warm-up steps. For the diffusion process, we use 200 steps adding noise to the series or reconstructing them. More details about implementation can be found in the Appendix E.

## 4.2 MAIN RESULTS

In this experiment, we evaluate our methods with 6 baselines in Table 1. Our method achieves the best performance in 10 out of the 12 datasets measuring with maximum mean discrepancy, which indicates that the time-series generated by our model conform to the original sequence better than other competitive methods. Considering all of the 12 datasets, our method averagely decreases the

Table 2: Results for ablation study. The best results of each line are **bold**. Window-wise alignment outperforms other methods with 15 best scores.

| Alignment Type | Window-wise Alignment | | Patch-wise Alignment | | Point-wise Alignment | |
|---|---|---|---|---|---|---|
| Metric | MMD | K-L | MMD | K-L | MMD | K-L |
| Electricity | $\mathbf{0.001}_{\pm 0.002}$ | $\mathbf{0.008}_{\pm 0.010}$ | $0.002_{\pm 0.005}$ | $0.011_{\pm 0.005}$ | $0.002_{\pm 0.004}$ | $0.030_{\pm 0.028}$ |
| Solar | $\mathbf{0.035}_{\pm 0.004}$ | $\mathbf{0.013}_{\pm 0.005}$ | $\mathbf{0.035}_{\pm 0.003}$ | $0.016_{\pm 0.007}$ | $\mathbf{0.035}_{\pm 0.003}$ | $0.017_{\pm 0.011}$ |
| Wind | $0.034_{\pm 0.014}$ | $0.202_{\pm 0.044}$ | $\mathbf{0.033}_{\pm 0.010}$ | $\mathbf{0.176}_{\pm 0.005}$ | $0.034_{\pm 0.010}$ | $0.186_{\pm 0.026}$ |
| Traffic | $\mathbf{0.071}_{\pm 0.005}$ | $\mathbf{0.011}_{\pm 0.002}$ | $0.109_{\pm 0.012}$ | $0.012_{\pm 0.010}$ | $0.073_{\pm 0.008}$ | $\mathbf{0.010}_{\pm 0.003}$ |
| Taxi | $\mathbf{0.085}_{\pm 0.010}$ | $\mathbf{0.005}_{\pm 0.003}$ | $0.195_{\pm 0.000}$ | $0.014_{\pm 0.000}$ | $0.088_{\pm 0.011}$ | $\mathbf{0.004}_{\pm 0.002}$ |
| Pedestrain | $0.040_{\pm 0.008}$ | $\mathbf{0.009}_{\pm 0.004}$ | $\mathbf{0.034}_{\pm 0.014}$ | $0.011_{\pm 0.135}$ | $0.041_{\pm 0.010}$ | $0.012_{\pm 0.009}$ |
| Air | $\mathbf{0.011}_{\pm 0.002}$ | $\mathbf{0.026}_{\pm 0.011}$ | $0.034_{\pm 0.001}$ | $0.045_{\pm 0.016}$ | $0.012_{\pm 0.001}$ | $0.028_{\pm 0.010}$ |
| Temperature | $0.230_{\pm 0.021}$ | $\mathbf{0.176}_{\pm 0.027}$ | $\mathbf{0.214}_{\pm 0.004}$ | $0.177_{\pm 0.009}$ | $0.217_{\pm 0.026}$ | $\mathbf{0.176}_{\pm 0.042}$ |
| Rain | $\mathbf{0.036}_{\pm 0.016}$ | $\mathbf{0.011}_{\pm 0.003}$ | $0.052_{\pm 0.027}$ | $0.039_{\pm 0.033}$ | $0.039_{\pm 0.024}$ | $\mathbf{0.011}_{\pm 0.004}$ |
| NN5 | $0.147_{\pm 0.008}$ | $\mathbf{0.045}_{\pm 0.007}$ | $\mathbf{0.146}_{\pm 0.015}$ | $0.080_{\pm 0.006}$ | $0.152_{\pm 0.008}$ | $0.048_{\pm 0.013}$ |
| Fred-MD | $\mathbf{0.002}_{\pm 0.001}$ | $0.201_{\pm 0.014}$ | $0.002_{\pm 0.005}$ | $\mathbf{0.199}_{\pm 0.003}$ | $\mathbf{0.002}_{\pm 0.001}$ | $0.218_{\pm 0.048}$ |
| Exchange | $\mathbf{0.137}_{\pm 0.012}$ | $1.621_{\pm 0.126}$ | $0.139_{\pm 0.011}$ | $\mathbf{1.595}_{\pm 0.023}$ | $0.139_{\pm 0.014}$ | $1.625_{\pm 0.115}$ |
| Count | **15** | | 8 | | 6 | |

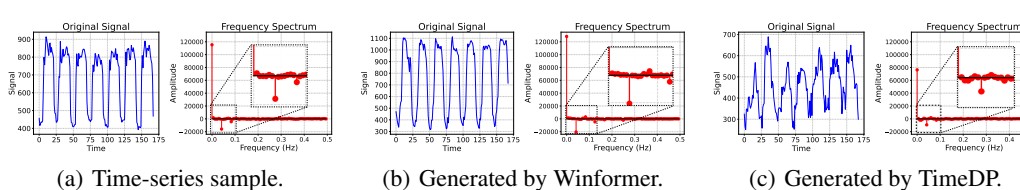

(a) Time-series sample.  (b) Generated by Winformer.  (c) Generated by TimeDP.

Figure 4: Visualization of generated series, comparing on original signal and frequency spectrum.

MMD with $10.67\%$ compared to TimeDP, which is the SOTA method of cross-domain generation. Besides, we can find that the Winformer demonstrates especially stronger ability on datasets with strong periodicity, such as solar and traffic. On the contrary, Winformer's ability lags slightly behind on datasets exhibiting a stronger tendency toward trendiness rather than periodicity, such as wind and temperature. This experimental phenomenon is also consistent with our understanding and analysis of window-to-window alignment. Since periodicity is widely present and plays a crucial role in real-world time-series datasets, our Winformer achieves superior performance across most datasets when evaluated against both KL and MMD metrics. For further analysis on periodicity capturing, we visualized an example of the generated series with both original signal and frequency spectrum as shown in Fig. 4. It's obvious that time-series signals generated by our method are more similar to the original data. By observing the frequency spectrum, we can infer that our window-to-window alignment method can extract more detailed periodic patterns than TimeDP, which probably explains why our method outperforms SOTA baselines.

### 4.3 ABLATION STUDY

We conduct ablation study with the results in Table 2. We evaluated the performance with **window-wise alignment**, which is proposed by us, **patch-wise alignment** and **point-wise alignment**. The window-wise alignment achieves the best performance on most of the datasets. And patch-wise alignment shows slightly advantages in dataset wind considering both KL and MMD, which is because the periodic features in this dataset is relatively weak. However, our model still has obvious superiority in most scenarios, because periodicity is widely present in real-world time-series data.

## 5 DISCUSSION

**Q1: How the Ample attention helps the diffusion model recovering the time-series patterns?**
We conduct experiments on a synthetic time-series data as Figure 5(a), and it contains sinusoidal signals with different cycle periods. The largest period is 50, and the smallest is 5. We evaluate

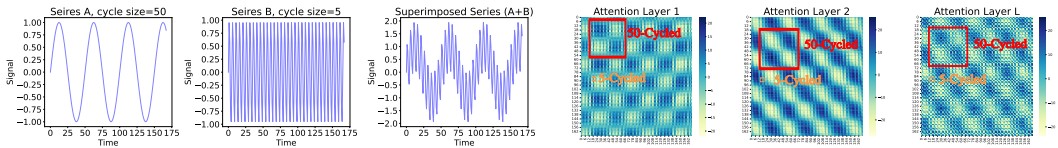

(a) The synthetic data contains different periodic signals.   (b) Winformer captures periodicity in synthetic data.

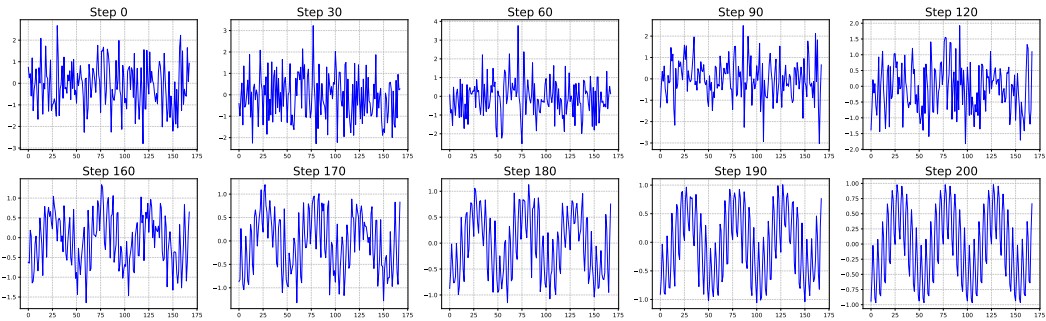

(c) The visualization of Winformer's denoising process on the synthetic data.

Figure 5: The detailed discussion and performance exploration on the synthetic data.

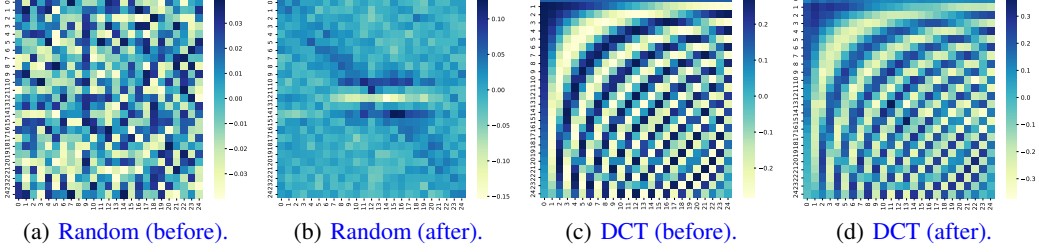

(a) Random (before).   (b) Random (after).   (c) DCT (before).   (d) DCT (after).

Figure 6: Visualization of the kernels for window-wise alignment, with random initialization or with DCT-based initialization. The visualization compares the kernel before and after training.

the proposed model on the synthetic data and draw the product score's visualization between input $Q$ and $K$ in Eq.(1). Due to space limit, we place the score matrix of the first two layers and the last one (more visualization figures can be found in Appendix C), where the darker area in heatmap indicates closer time-series interactions. The swapped color stripes reveals distinct two kinds of cycle patterns: the smaller orange grid is 5-cycled and the bigger red one is 50-cycled. Thus, we could leverage the Ample attention to capture the periodic interactions for better recovering.

**Q2: How does the Winformer architecture utilize the periodic information during the denoising process?** We visualized the denoising process in Figure 5(c), and the longer cycle firstly emerges then the rests. Taking a concrete example at step 140, it presents a rough cycle with period 50, then the cycle period shrinks to 5 at step 180. The Winformer appears to first identify major periodic patterns that have wider impact, then incorporate more detailed periodic patterns. This adaptive nature enables Winformer to have generalized window alignment capability in denoising process, particularly when applied to the cross-domain cases.

**Q3: How does the kernel changes during the training process?** We have explored the changes of the kernel. Specifically, we tested two kernel initialization methods. One using random initialization, and the other using our proposed DCT-based initialization. The visualizations for kernel with random initialization are shown in Fig. 6(a) and Fig. 6(b), which are the kernel weights before and after training. It can be observed that the values in the weight matrix tend to be evenly distributed. This means the convolutional layer may reduce during the training process, and it gradually trans-

formed as 'kernel' of the average pooling layer, whose weights for each position are very similar. The visualizations for kernel initialized with DCT are shown in Fig. 6(c) and Fig. 6(d). It can be observed that the weights do not change significantly, which is because the DCT basis already has a good effect on capturing periodicity.

# 6 RELATED WORK

## 6.1 TIME-SERIES GENERATION

Existing time-series generation models mainly consist of GAN-based models, VAE-based models and diffusion-based models. GAN-based models (Yoon et al., 2019; Jeon et al., 2022) apply adversarial networks consisting of generators and discriminators without an explicit probability distribution assumption. However, the adversarial process shows poor training stability. VAE-based models (Desai et al., 2021; Lee et al., 2023) can achieve training stability with clear optimization target. But VAE models show limited generation quality and diversity. Recent diffusion models (Huang et al., 2025; Ge et al., 2025) can flexibly capture complex patterns within time-series data with superior efficiency and effectiveness, holding significant advantages in time-series generation.

## 6.2 TRANSFORMER-BASED DIFFUSION MODELS

Transformer-based diffusion models show a more powerful effect in various time-series tasks, including forecasting (Feng et al., 2024), anomaly detection (Wang & Li, 2025) and generation (Peebles & Xie, 2023). In addition to these domain-specific models, there are also foundational time-series models for general purpose such as TimeDiT (Cao et al., 2025). Moreover, transformer-based models are beneficial for adapting to different modalities, leading to a unified diffusion model. For example, T2S (Ge et al., 2025) combines the text modality and time-series modality with the diffusion transformer model. In light of these advantages, we further investigate the performance of transformer-based models in cross-domain tasks.

# 7 CONCLUSION

We address the challenge of periodicity alignment for cross-domain time-series generation by reforming the attention mechanism in window-wise alignment. Specifically, we propose Winformer, a diffusion framework built on window-wise modeling, which replaces pairwise similarity in vanilla attention with window-to-window comparison via time-frequency analysis. With theoretical deduction, the window-wise alignment is reducible to learnable convolutions, enabling effective implementation. Winformer show advanced performance by achieving an average 10.67% MMD improvement over SOTA baselines on 12 real-world datasets. Visualization analysis and ablation studies verified the effectiveness of our window-wise alignment. Future works will focus on better conditioning, multi-modal integration, and diffusion efficiency optimization.

## REPRODUCIBILITY STATEMENT

To ensure the reproducibility of our work, we include necessary information in the main text, appendix, and supplementary materials. **Codes and Datasets**: Our codes are upload anonymously to a secure repository [1] with an Quick-Start document. The datasets are open-sourced by previous works[2]. **Experiment Details:** As mentioned in Section 4.1 of the main text, we provide an overview of the experimental settings. More detailed information can be found in Appendix E, including detailed introduction of metrics, datasets, hyper-parameters and other implementation details. **Theoretical Derivations:** We have included the theoretical derivations in Section 3 with some theoretical backgrounds in Appendix G and the computational costs in Appendix B. **Results:** We provide main results, ablation results in Section 4. More results, including hyper-parameter studies and visualizations, are provided in Appendix A and Appendix C. We hope these resources collectively enable researchers to replicate our experimental setup, theoretical verifications, and result analyzes.

---

[1] https://anonymous.4open.science/r/Winformer-anonymize/

[2] https://huggingface.co/datasets/YukhoW/TimeDP-Data/blob/main/TimeDP-Data.zip

ETHICS STATEMENT

All authors have read, acknowledged, and fully adhered to the ICLR Code of Ethics. Our work prioritizes public good. We use open-sourced datasets with no sensitive data, checked for biases to avoid harm, and report results honestly. No conflicts of interest exist. We uphold the code's aim of guiding ethical research that advances knowledge and well-being.

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

## A  HYPER-PARAMETER STUDY

In this experiment, we evaluated the performance with different kernel types and sizes. The results are shown in Table 3, and the best model, which adapts a 25-size Conv2d layer as the aligning kernel, whose results are also selected to be shown in Table 1 competing with SOTA baselines.

**For kernel types**, we apply two types of kernels. The Conv2d kernel is implemented with a 2D convolution neural network, whose kernel is initialized with the DCT basis. The Avgpool2d kernel is implemented with a 2D average pooling operator. The results show that Conv2d performs better than Avgpool2d, and both operators show effectiveness compared to the model without any kernel. We can find that DCT kernel performs better than Avg pooling because DCT kernel could capture more periodicity information, while DCT achieves 8+8+9=25 best counts and Avg achieves 4+9+5=18 best counts. Thus, in this paper, we mainly consider the DCT kernels. By the way, Avg Pooling is more efficient, so it's more suitable with limited calculation consumption. **For kernel sizes**, we select three sizes, including 7, 13 and 25, which are odd numbers to facilitate the convolution operator. The results show that, for Conv2d kernel, model with the size of 25 performs best because it contains the cycle up to 24, which is a common cycle for time-series data. The optimal 25-size kernel in the paper works because it includes typical cycles (4, 6, 12, 24) without overcomplicating computation.

Table 3: Results for hyper-parameters study . Best results of each line are **bold**. The best model uses 2D Convolution layer with the kernel size of 25, and it achieves best performance on 9 lines in total, which is also selected as our final model shown in Table 1.

| # Kernel Type | | Window-wise align by Conv2d kernel | | | Window-wise align by Avgpool2d kernel | | |
|---|---|---|---|---|---|---|---|
| # Kernel Size | | 7 | 13 | 25 | 7 | 13 | 25 |
| Maximum Mean Discrepancy | Electricity | $\mathbf{0.001}_{\pm 0.001}$ | $0.002_{\pm 0.003}$ | $\mathbf{0.001}_{\pm 0.002}$ | $0.002_{\pm 0.003}$ | $0.002_{\pm 0.003}$ | $0.002_{\pm 0.003}$ |
| | Solar | $0.035_{\pm 0.001}$ | $0.035_{\pm 0.002}$ | $0.035_{\pm 0.004}$ | $\mathbf{0.033}_{\pm 0.005}$ | $\mathbf{0.033}_{\pm 0.004}$ | $\mathbf{0.033}_{\pm 0.004}$ |
| | Wind | $0.035_{\pm 0.001}$ | $\mathbf{0.033}_{\pm 0.011}$ | $0.034_{\pm 0.014}$ | $\mathbf{0.033}_{\pm 0.010}$ | $\mathbf{0.033}_{\pm 0.010}$ | $0.037_{\pm 0.013}$ |
| | Traffic | $0.074_{\pm 0.007}$ | $0.074_{\pm 0.003}$ | $\mathbf{0.071}_{\pm 0.005}$ | $0.072_{\pm 0.003}$ | $\mathbf{0.071}_{\pm 0.003}$ | $0.074_{\pm 0.005}$ |
| | Taxi | $\mathbf{0.080}_{\pm 0.015}$ | $0.086_{\pm 0.012}$ | $0.085_{\pm 0.010}$ | $0.084_{\pm 0.013}$ | $0.083_{\pm 0.012}$ | $0.082_{\pm 0.011}$ |
| | Pedestrain | $\mathbf{0.040}_{\pm 0.011}$ | $0.041_{\pm 0.010}$ | $\mathbf{0.040}_{\pm 0.008}$ | $0.042_{\pm 0.009}$ | $0.041_{\pm 0.010}$ | $\mathbf{0.040}_{\pm 0.011}$ |
| | Air | $0.014_{\pm 0.004}$ | $0.013_{\pm 0.001}$ | $\mathbf{0.011}_{\pm 0.002}$ | $0.013_{\pm 0.001}$ | $0.013_{\pm 0.001}$ | $0.012_{\pm 0.002}$ |
| | Temperature | $0.226_{\pm 0.022}$ | $\mathbf{0.219}_{\pm 0.027}$ | $0.230_{\pm 0.021}$ | $0.220_{\pm 0.026}$ | $\mathbf{0.219}_{\pm 0.024}$ | $0.229_{\pm 0.023}$ |
| | Rain | $\mathbf{0.033}_{\pm 0.012}$ | $0.050_{\pm 0.046}$ | $0.036_{\pm 0.016}$ | $0.038_{\pm 0.024}$ | $0.037_{\pm 0.019}$ | $0.036_{\pm 0.016}$ |
| | NN5 | $0.149_{\pm 0.005}$ | $0.151_{\pm 0.007}$ | $\mathbf{0.147}_{\pm 0.008}$ | $0.153_{\pm 0.006}$ | $0.154_{\pm 0.006}$ | $0.154_{\pm 0.007}$ |
| | Fred-MD | $\mathbf{0.002}_{\pm 0.001}$ | $\mathbf{0.002}_{\pm 0.001}$ | $\mathbf{0.002}_{\pm 0.001}$ | $\mathbf{0.002}_{\pm 0.001}$ | $\mathbf{0.002}_{\pm 0.001}$ | $\mathbf{0.002}_{\pm 0.001}$ |
| | Exchange | $\mathbf{0.136}_{\pm 0.012}$ | $0.139_{\pm 0.015}$ | $0.137_{\pm 0.012}$ | $0.140_{\pm 0.015}$ | $0.139_{\pm 0.014}$ | $0.138_{\pm 0.014}$ |
| K-L Divergence | Electricity | $0.019_{\pm 0.030}$ | $0.020_{\pm 0.018}$ | $\mathbf{0.008}_{\pm 0.010}$ | $0.033_{\pm 0.044}$ | $0.023_{\pm 0.021}$ | $0.014_{\pm 0.010}$ |
| | Solar | $0.021_{\pm 0.013}$ | $\mathbf{0.012}_{\pm 0.004}$ | $0.013_{\pm 0.005}$ | $0.015_{\pm 0.008}$ | $0.015_{\pm 0.009}$ | $0.016_{\pm 0.012}$ |
| | Wind | $0.201_{\pm 0.040}$ | $0.186_{\pm 0.030}$ | $0.202_{\pm 0.044}$ | $\mathbf{0.180}_{\pm 0.026}$ | $0.182_{\pm 0.026}$ | $0.203_{\pm 0.040}$ |
| | Traffic | $0.013_{\pm 0.007}$ | $\mathbf{0.009}_{\pm 0.003}$ | $0.011_{\pm 0.002}$ | $0.010_{\pm 0.003}$ | $\mathbf{0.009}_{\pm 0.002}$ | $0.011_{\pm 0.005}$ |
| | Taxi | $0.006_{\pm 0.004}$ | $\mathbf{0.004}_{\pm 0.003}$ | $0.005_{\pm 0.003}$ | $0.005_{\pm 0.002}$ | $\mathbf{0.004}_{\pm 0.001}$ | $\mathbf{0.004}_{\pm 0.001}$ |
| | Pedestrain | $\mathbf{0.009}_{\pm 0.004}$ | $0.011_{\pm 0.005}$ | $\mathbf{0.009}_{\pm 0.004}$ | $0.012_{\pm 0.004}$ | $0.012_{\pm 0.004}$ | $0.010_{\pm 0.004}$ |
| | Air | $0.031_{\pm 0.012}$ | $0.025_{\pm 0.009}$ | $0.026_{\pm 0.011}$ | $0.028_{\pm 0.010}$ | $0.027_{\pm 0.009}$ | $\mathbf{0.023}_{\pm 0.008}$ |
| | Temperature | $\mathbf{0.173}_{\pm 0.027}$ | $0.174_{\pm 0.039}$ | $0.176_{\pm 0.027}$ | $0.178_{\pm 0.036}$ | $0.181_{\pm 0.031}$ | $0.186_{\pm 0.030}$ |
| | Rain | $0.014_{\pm 0.008}$ | $\mathbf{0.009}_{\pm 0.004}$ | $0.011_{\pm 0.003}$ | $0.011_{\pm 0.004}$ | $0.015_{\pm 0.008}$ | $0.012_{\pm 0.004}$ |
| | NN5 | $0.048_{\pm 0.009}$ | $0.049_{\pm 0.018}$ | $\mathbf{0.045}_{\pm 0.007}$ | $0.047_{\pm 0.012}$ | $0.054_{\pm 0.018}$ | $0.053_{\pm 0.026}$ |
| | Fred-MD | $0.248_{\pm 0.109}$ | $\mathbf{0.197}_{\pm 0.028}$ | $0.201_{\pm 0.014}$ | $0.201_{\pm 0.020}$ | $\mathbf{0.197}_{\pm 0.020}$ | $0.204_{\pm 0.024}$ |
| | Exchange | $1.629_{\pm 0.166}$ | $1.594_{\pm 0.151}$ | $1.621_{\pm 0.126}$ | $1.610_{\pm 0.159}$ | $\mathbf{1.590}_{\pm 0.155}$ | $1.635_{\pm 0.125}$ |
| Count | | 8 | 8 | **9** | 4 | 9 | 5 |

## B  COMPUTATION COSTS

In this section, we try to analyze the computational costs of the window-to-window aligned attention. We assume the length of time-series if $L_x$, and the dimension is $D$. The computation complexity of the calculation for similarity scores, which is the dot-product of $\mathbf{Q}$ and $\mathbf{K}$, is $\mathcal{O}(L_x^2 D)$. If the windows size is set to $s$, which means to compute with convolutional network with kernel size $s$,

the computational complexity is $\mathcal{O}(s^2)$. We travel through all the time points of $\mathbf{Q}$ and $\mathbf{K}$, the total computation of $\mathbf{S}$ becomes $\mathcal{O}(s^2 \cdot L_x^2 D)$. We also tested the actual costs of the window-wise alignment as shown in Table 4. The training speed is to test how fast a denoising model can denoising a batch of data with gradient backpropagation during the training process. The sampling speed evaluates how fast a denoising model can make a prediction during the testing process. Although our window-wise alignment increases the complexity, in practical experiments, the actual computation consumption is acceptable with slight increase comparing with point-wise alignment because the calculation of convolutional neural operators has been optimized by machine learning frameworks.

Table 4: Computational costs for different kinds of alignment method. We evaluate the average speed of training and sampling of each batch (batch size=128).

| Model | Kernel Size | Train Speed (batch/s) | Sample Speed (batch/s) |
|---|---|---|---|
| Point-wise Alignment | - | 2.92 | 10.57 |
| Winformer (Window-wise Alignment) | 7 | 2.58 | 9.69 |
| | 13 | 2.33 | 9.13 |
| | 25 | 2.30 | 8.95 |

## C  VISUALIZATIONS AND DISCUSSIONS

We provide extended visualized figures on real-world datasets, including the visualization of the attention similarity matrix, the visualization of the denoising process and the visualization of the generated time-series. We further discuss about these visualized figures in the following subsections.

### C.1  VISUALIZATION OF THE ATTENTION

In this subsection, we want to explore how can the Ample attention help the diffusion model in recovering the time-series patterns. We conduct experiments on the synthetic datasets to discover whether the model utilize the periodic pattern in denoising, which inspire us to enhance the periodic features with the Ample attention. We also verify the effectiveness on periodic enhancement by visualizing the similarity score before and after window-wise alignment.

#### C.1.1  VISUALIZATION ON SYNTHETIC DATA

We conduct the experiments on the synthetic datasets, which contains various periodic patterns as defined in Eq. 18. By observing the phenomena on the synthetic dataset, we can explore how the denoising model works with periodic capturing. We visualized the similarity score of the attention mechanism in Figure 7. In the heatmap of the score, the darker the color represents the deeper similarity of the time-series points. These visualized images show periodic cycles with repetitive square-like patterns. From layer 1 to Layer $L$ ($L = 6$), we can find that the patterns may shift with constant steps. The phenomena enlighten us that the periodic features are essential for transformer-based time-series denoising models.

#### C.1.2  VISUALIZATION ON REAL-WORLD DATA

We also visualized the similarity score matrices of the self-attention mechanism during the model's denoising process on a portion of real-world data, as shown in Figure 8. This figure corresponds to the taxi dataset, which contains strong periodic patterns. The three subgraphs represent the kernel for window-wise alignment, the similarity score matrix before alignment, and the similarity matrix after window-wise alignment. The time-series data in the selected real-world datasets exhibit good periodicity, so their similarity matrices show a grid pattern. After window-wise alignment, we notice that the long-range grids become more distinct, indicating the window-wise alignment fuse features inside the observation window. Thus, our method can enhance the periodic features.

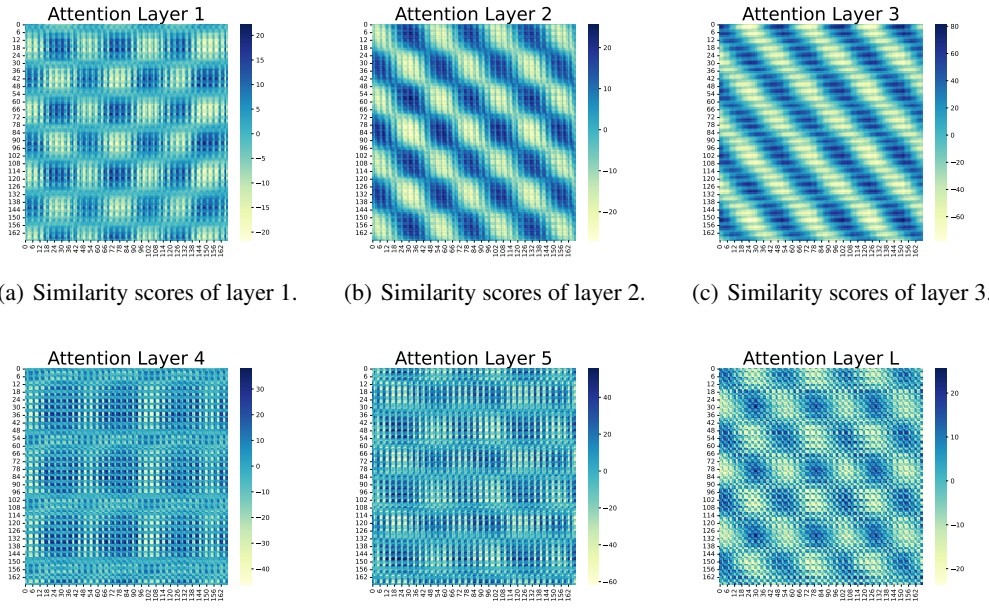

(a) Similarity scores of layer 1.  (b) Similarity scores of layer 2.  (c) Similarity scores of layer 3.

(d) Similarity scores of layer 4.  (e) The similarity scores of layer 5.  (f) The similarity scores of layer 6.

Figure 7: Visualization of the similarity matrices for synthetic data.

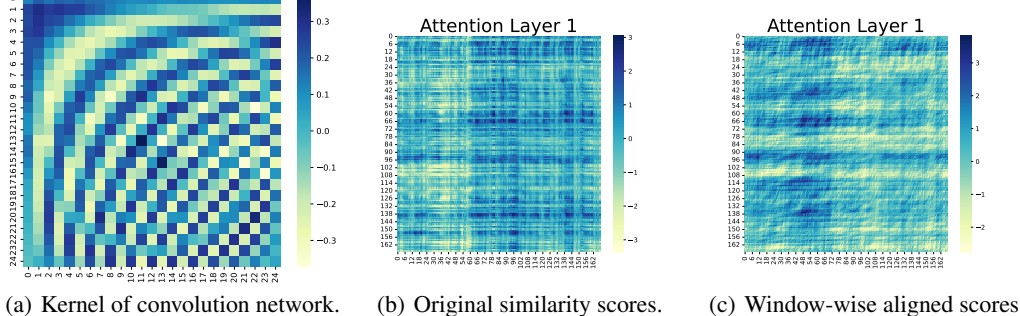

(a) Kernel of convolution network.  (b) Original similarity scores.  (c) Window-wise aligned scores.

Figure 8: Visualization of the kernel for window-wise alignment, the similarity matrix before alignment, and the similarity matrix after window-wise alignment for dataset Taxi.

## C.2 VISUALIZATION FOR ALIGNMENT COMPARISON

To explore the differences in the effectiveness of the three kinds alignments, namely point-wise, patch-wise and window-wise, in extracting complex periodicity, we conducted result visualization on taxi dataset. Taxi dataset contains complex periodicity structure, as not only presents the daily cycle but also includes the detailed periodic changes of morning and evening peaks. We visualized the series generated with point-wise, patch-wise and window-wise alignment in Fig. 9. We can find that, window-wise method could capture the periodicity better than other methods, by well establishing periodicity structures.

## C.3 VISUALIZATION OF THE GENERATED TIME-SERIES

We visualized some time-series data generated by the Winformer, as shown in Fig. 10. The solar dataset contains a simple daily cycle. These generated series, containing obvious periodicity, indicates that the Winformer can effectively capture the periodic features. This confirms that our window-wise alignment can enhance the capturing of periodicity.

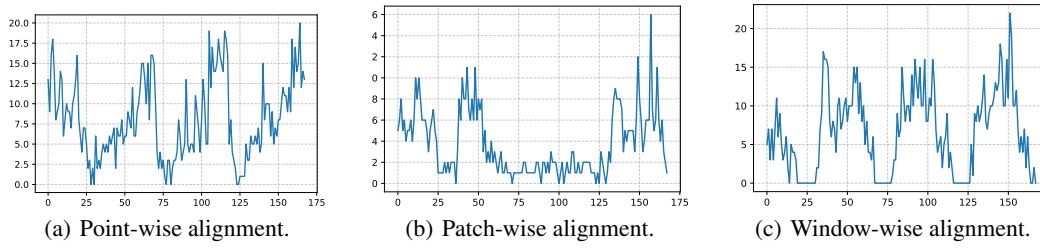

(a) Point-wise alignment.      (b) Patch-wise alignment.      (c) Window-wise alignment.

Figure 9: Visualization of the series generated by point-wise, patch-wise and window-wise alignment for dataset Taxi.

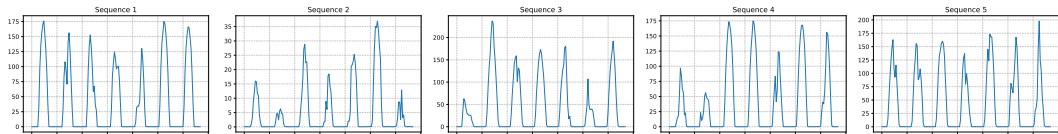

Figure 10: The generated time-series data for dataset Solar.

## C.4 VISUALIZATION OF THE DENOISING PROCESS

To further explore the denoising process in time-series generation tasks, we visualized the sequences during the denoising process on real-world dataset, as shown in Figure 11. The Solar datasets contains obvious daily cycles. By observing these images, we can find that in the denoising process of time-series data, long-range periodic information is captured first and displayed initially. Other detailed information will be revealed in later steps of denoising. Therefore, we propose the hypothesis that in the view of denoising model, the basic components for time-series data are frequency features instead of temporal points. That's explain and verify why the Winformer can achieve better performance against SOTA baselines.

## D DOMAIN DISCREPANCY

We add a supplementary evaluation on domain discrepancy. Specifically, we evaluate the KL-divergence between different domains. The results are shown in Fig 12. To facilitate observation, all data in the figure are logarithmically transformed (log-transformed). For most datasets, the KL divergence between the generated data and the real data in their respective domains is the lowest, indicating that the domain-specific features are well learned.

## E EXPERIMENT DETAILS

We further report the detailed settings of the experiments, including information of the measurement, datasets and implementation.

### E.1 METRICS AND MEASUREMENT

In this section, we describe the metrics and other settings which related to the measurement of the model's performance. First, we formulate the key metrics evaluating the performance of the time-series generation. Then we describe the detail settings for repeated experiments and how we get the reported results with error bounds.

#### E.1.1 FORMULATED METRICS

In this paper, we adopt two metrics in measuring the quality of the generated time-series data. Firstly, we define a real time-series data with $L$ length and $D$ channels as $\mathbf{X} = [\mathbf{X}_1, \mathbf{X}_2, ..., \mathbf{X}_L] \in$

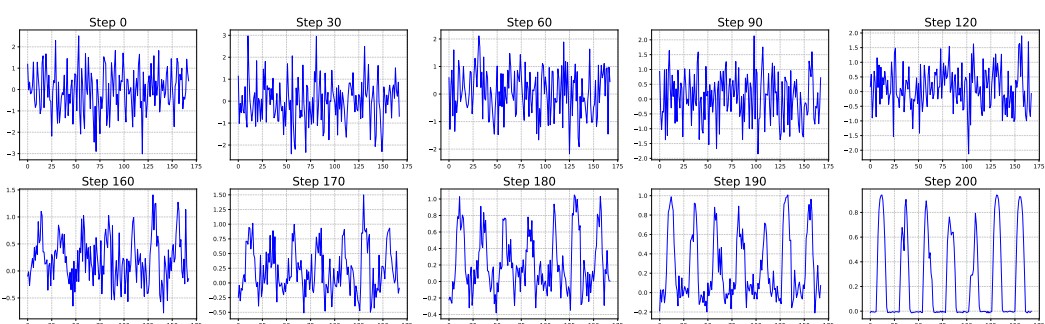

Figure 11: Denoising Process for the dataset of Solar.

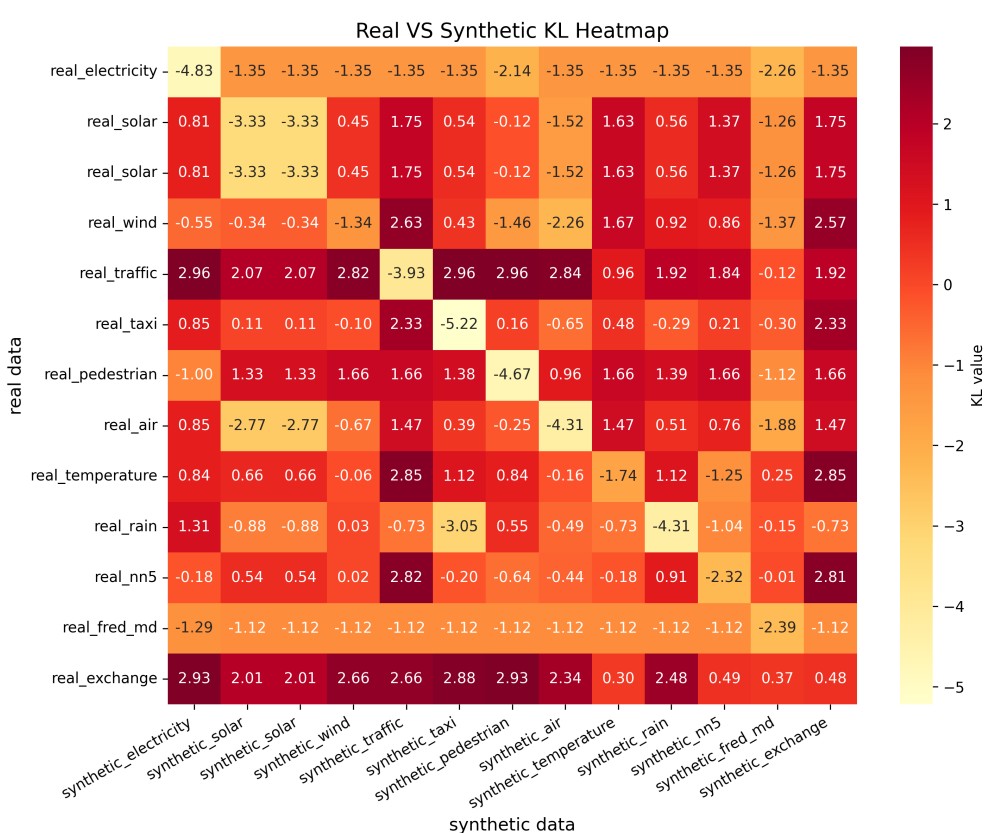

Figure 12: The domain discrepancy among datasets.

$\mathbb{R}^{L \times D}$, and the synthetic data is $\hat{\mathbf{X}} = [\hat{\mathbf{X}}_1, \hat{\mathbf{X}}_2, ..., \hat{\mathbf{X}}_L] \in \mathbb{R}^{L \times D}$. Then we formulate the metrics of Maximum Mean Discrepancy and Kullback-Leibler Divergence as follow.

**(1) Maximum Mean Discrepancy (MMD).** The MMD is a distribution similarity evaluation method. Specifically, we transform the time-series data into a high-dimension space by $\Phi(\cdot, \cdot)$. Then we calculates the average of the results obtained by the kernel to get the MMD, which is formulated as follow:

$$L_{MMD} = \frac{\sum_{i=1}^{N} \Phi(\mathbf{X}_i, \mathbf{X}_i)}{N} + \frac{\sum_{i=1}^{N} \Phi(\hat{\mathbf{X}}_i, \hat{\mathbf{X}}_i)}{N} - 2\frac{\sum_{i=1}^{N} \Phi(\mathbf{X}_i, \hat{\mathbf{X}}_i)}{N} , \quad (14)$$

Table 5: Details for the real-world datasets.

| Domain | Dataset | Variables | Sampling interval | Description | Source |
|---|---|---|---|---|---|
| Energy | Electricity | 321 | 1 hour | the electricity consumption | UCI |
| | Solar | 137 | 1 hour | the solar power production | State of Alabama |
| | Wind | 1 | 4 second | the wind power production | AEMO |
| Weather | Air | 270 | 1 hour | the air quality levels | KDDCup2018 |
| | Temperature | 422 | 1 day | the temperature observations | Australia |
| | Rain | 422 | 1 day | the rain forecast | Australia |
| Transportation | Traffic | 963 | 1 hour | the occupancy rate of car lanes | San Francisco bay |
| | Taxi | 1214 | 30 minutes | the taxi rides | New York |
| | Pedestrian | 1 | 1 hour | the pedestrian counts | Melbourne city |
| Economic | NN5 | 111 | 1 day | the cash withdrawals from ATMs | UK |
| | Fred-MD | 107 | 1 month | the macro-economic indicators | Federal Reserve Bank |
| | Exchange | 8 | 1 day | the exchange rate | Reports of 8 countries |

where $\Phi(\cdot, \cdot)$ is implemented by the radial basis function kernel.

**(2) Kullback-Leibler Divergence(K-L).** The K-L is a common metric measuring the similarity between real and synthetic data.

$$L_{K-L} = \sum_{i=1}^{K} P(\mathbf{X}) \log \left( \frac{P(\mathbf{X})}{Q(\hat{\mathbf{X}})} \right) \quad , \tag{15}$$

where $P(\cdot)$ and $Q(\cdot)$ are mapping functions to obtain the distribution of the data by reforming the data into histogram, which includes $K$ indexes in total.

### E.1.2 REPEATED EXPERIMENTS

All experiments are repeated five times with random seeds ranging from 2021 to 2025. To demonstrate the comprehensive effect of the model, we report the average results with their standard deviation of the repeated experiments.

### E.2 DATASETS

We conducted the time-series generation experiment on 12 real-world datasets and a synthetic dataset. The description of the datasets are as follow.

### E.2.1 REAL-WORLD DATASETS

In this paper, we conduct the experiments following the setting of the TimeDP Huang et al. (2025). The experiments contain 12 real-world datasets from four domains, including energy, economic, weather and transportation. The pre-processed datasets are open-sourced by TimeDP[3]. We list the details of the datasets in Table 5.

### E.2.2 SYNTHETIC DATASET

The synthetic datasets contains time-series data which are the combination of the sinusoidal signal with different periods. Specifically, let $\mathbf{X}_1(t)$ be the sinusoidal signal with a cycle of 50, which is defined as follow:

$$\mathbf{X}_1(t) = sin(\frac{2\pi}{50}t) \quad , \tag{16}$$

and $\mathbf{X}_2(t)$ be the sinusoidal signal with a cycle of 5 defined as:

$$\mathbf{X}_2(t) = sin(\frac{2\pi}{5}t) \quad . \tag{17}$$

---

[3]https://huggingface.co/datasets/YukhoW/TimeDP-Data/blob/main/TimeDP-Data.zip

Table 6: Hyper-parameters for the Winformer.

| Catergory | Module | Name | Value |
|---|---|---|---|
| Architecture of Winformer | Conditioning block | Dimension | 32 |
| | | Channel | 1 |
| | | Latent | 1 |
| | Hybrid encoder block | Channel | 1 |
| | | Hidden size | 512 |
| | | Vanilla attention heads | 8 |
| | | Ample attention heads | 8 |
| | | MLP ratio | 4.0 |
| | | Kernel type | Conv2d |
| | | Kernel initialization | discrete cosine transform (DCT) |
| | | Kernel size | 25 |
| | DiT encoder block | Hybrid encoder block | 6 |
| | | Channel | 1 |
| | | Vanilla attention heads | 16 |
| | | MLP ratio | 4.0 |
| | Forward/Reverse Process | Noising/Denoising steps | 200 |
| | | Loss type | L1 loss |
| | Trainer | Batch size | 128 |
| | | Learning rate | 5e-5 |
| | | Train steps | 50, 000 |

Then, we can obtain the synthetic data by superposing these two series as

$$\mathbf{X}(t) = \mathbf{X}_1(t) + \mathbf{X}_2(t) \quad . \tag{18}$$

Thus, we get the synthetic data $\mathbf{X}(t)$ which contains two types of cycles. We sample 20000 time steps and split the sequences into 168-length time series data. By conducting denoising method on the synthetic data, we can explore how the Ample attention and the Winformer helps in frequency capturing and cross-domain time-series generation. From visualization of the denoising process, we can find that the model firstly recovers the large-range periodic patterns. This motivates us to enhance the periodic pattern capturing by window-wise alignment, which is exactly our method does.

### E.3 IMPLEMENTATION SETTINGS

In this section, we report the detailed implementation, including hyper-parameters and environments.

#### E.3.1 HYPER PARAMETERS

Our method, namely Winformer is a transformer-based denoising model, consisting of several encoder blocks and a conditioning block. We have listed the detailed hyper-parameters of each component, which are shown in Table 6.

#### E.3.2 HARD-WARES AND ENVIRONMENTS

The experiments are conducted with a NIVDIA V100 GPU, with 32GB memory. Our model relies on public environment libraries, including CUDA, Python, PyTorch, etc. The framework is based on the source code of TimeDP[4]. Our codes are reported in the supplementary material. More detailed environment and installation tips are reported in the ReadMe file in codes folder.

---

[4]https://github.com/microsoft/TimeCraft/tree/main/TimeDP

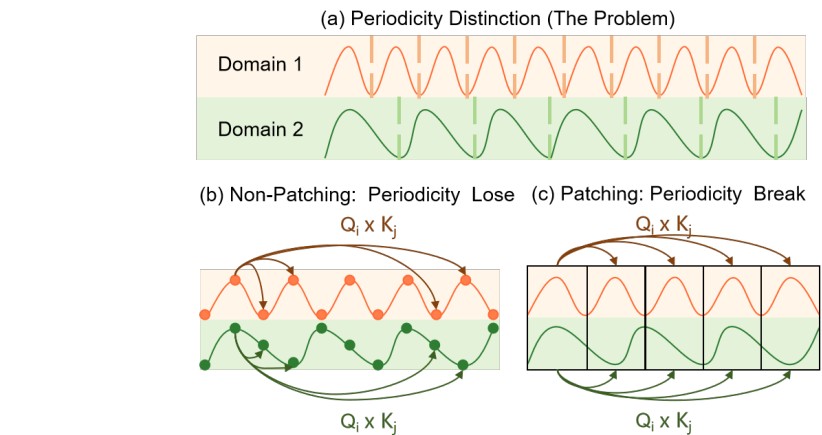

Figure 13: The observation for cross-domain time-series generation. For non-patching (point-wise) methods, the model understands each point relatively independently and lacks the ability to capture the correlation between adjacent time steps, which is detrimental to the acquisition of pattern structures. For patching methods, with information isolated and compressed at equal intervals, fixed windows will disrupt the periods of some sequences.

# F   TASK INTRODUCTION

## F.1   GENERATION VS. FORECASTING

Forecasting tasks emphasize on detailed perception for instances, while generation tasks requires global structure awareness for distributions. Forecasting and imputation tasks focus on estimating the series to the groundtruth under know observations in every single instances, but generation is to estimate the global distribution to the whole dataset. Generation tasks impose stronger requirements on more widely existing structures while downplaying the unique characteristics of individual samples. That's why we select the generation tasks to explore how could the model learn the periodicity structure better.

## F.2   OBSERVATIONS FOR CROSS-DOMAIN GENERATION

Cross-domain generation requires models to learn and transfer periodic patterns across diverse domains. As illustrated in Fig. 13(a), series from domain 1 and domain 2 show different periodicity, which is challenging for existing point-wise and patch-wise methods. Capturing the domain-specific periodicity directly tests the architecture's ability to generalize periodicity.

## F.3   THE DISCOVERY OF THE ADAPTIVE WINDOW-WISE ARCHITECTURE

The design of Winformer's adaptive window-wise architecture stems from a key observation, that is the limitations of exsting point-wise and patch-wise paradigms. Traditional point-wise attention struggles with complex dependencies and fails to model cyclic structures, as it only captures point-to-point interactions. As illustraed in Fig. 13(b), it's difficult to identify repetitive structures when time-series are numerically sensitive in continues space. Patch-wise methods fragment time-series into fixed segments, breaking evolving periodic patterns and requiring additional distribution restoration steps . Especially for cross-domain generation, models may process time-series with different windows, patching with a fixed segments could not be suitable for every domain, as illustrated in Fig. 13(c) . These flaws motivated us to rethink the "processing unit" of attention: Could we shift the discrete points or patches to continuous windows that naturally encapsulate periodic cycles? Compared to the point-wise methods, the window-wise methods can summary local features and simplified attention's work. Compared to the patch-wise mwthods, the window alignment is sliding, which allows to adapt different domains, as for every domain there is expected to have partial of the windows can contain the proper periodicity features. With these observations, we start for theoretical analysis to explore how to capture the periodicity with proper approaches.

## G  THEORETICAL BACKGROUNDS

In this section, we supplement theoretical backgrounds to facilitate presentation of the motivation and the design concepts. With these further explanation, we can better understand the underlying reasons and deduction process.

### G.1  THE FOURIER TRANSFORM

The Fourier Transform Duhamel & Vetterli (1990) is a mathematical operator that converts signal $\mathbf{x}$ to its frequency domain representation $\widetilde{\mathbf{x}}$. It is defined as a function $\mathcal{F}$ as:

$$\mathcal{F}\{u\}(\xi) = \int_{-\infty}^{+\infty} u(x)e^{-2\pi jx\xi}dx \quad . \tag{19}$$

The $u(x)$ is the input function to signal $\mathbf{x}$'s formulation. Then, we set the integral results as individual spectral components for different frequency $\xi$.

### G.2  CALCULATING SIMILARITY SCORE IN FREQUENCY DOMAIN

The attention mechanism, which calculates as follow:

$$\mathcal{A} = \text{Softmax}(\frac{\mathbf{Q}\mathbf{K}^{\mathrm{T}}}{\sqrt{d}})\mathbf{V} \quad , \tag{20}$$

where $\mathbf{Q}$ and $\mathbf{K}$ are mapped features from input $x$. To calculate the similarity score $\mathbf{S} = \mathbf{Q}\mathbf{K}^{\mathrm{T}}$ in the frequency domain, we can directly transform the input data with Fourier transform and calculate the score using the Fourier components. However, such a process is time-consuming and inflexible and with Fourier operators. For a learning-based model, we prefer a learning surrogator for this operation, which can be adaptively parameterized cooperating with the model. Thus, we need to further explore the simplified calculation of the score $\mathbf{S}$.

### G.3  CONVOLUTION THEOREM

To further explore the simplified calculation of the score, we have to consider how to transform the dot-product operator in frequency domain. As the convolution theorem McGillem & Cooper (1991) sates that:

$$\mathcal{F}\{u * v\} = \mathcal{F}\{u\} \cdot \mathcal{F}\{v\} \quad , \tag{21}$$

where $*$ is the convolution and the operator $\cdot$ represents the point-wise multiplication. The above derivations also apply for the discrete Fourier transform (DFT). Thus, there is the possibility of transforming the dot-production in the frequency domain into the convolution in the temporal domain. As a result, we can directly calculate the score $\mathbf{S}$ with convolution network, in the place of a series of the calculation for frequency transformation.

### G.4  PARSEVAL'S THEOREM

For discrete time signals, we consider a time-series data $x$ with length $n$. The Parseval's Theorem states:

$$\sum_{n=0}^{N-1} |x[n]|^2 = \frac{1}{N} \sum_{k=0}^{N-1} |X[k]|^2, \tag{22}$$

where $X[k]$ is the discrete Fourier transform (DFT) of $x[n]$. The Parseval's Theorem enables us to transfer correlation of frequency domain into that to time domain. Thus, we can adapts the deduction of Eq. 10 in the attention mechanism, leading to the final format of the Ample attention shown in Eq. 11.

## H  FUTURE WORKS

For cross-domain time-series generation, an important issue is to reduce the reliance on time-series samples with more powerful conditioning network. Besides, how to integrating time-series with

other modalities, such as text-to-series and image-to-series generation, is another underlying problem. Finally, the efficiency issue of the diffusion model also needs further exploration.

# I  THE USE OF LARGE LANGUAGE MODELS (LLMs)

In this work, we utilized large language models (LLMs) as an assist tool to aid or polish writing, and their roles did not rise to the level of a contributor. Specifically, the LLMs were used for limited purposes, which is to help refine the clarity and coherence of draft paragraphs in the main text by suggesting alternative phrasings, and validated by all human authors to ensure scientific accuracy and alignment with the research findings.

