# OpenReview forum: "Winformer: Transcending pairwise similarity for time-series generation"
_ICLR.cc/2026/Conference — Submitted to ICLR 2026_

### Official Review · Reviewer_z9c8 · 2025-10-28

**Soundness:** 3
**Presentation:** 3
**Contribution:** 2
**Rating:** 6
**Confidence:** 3

**Summary:**

This paper focus on time series generation problem. By leveraging the Transformer and diffusion framework, a model termed Winformer is proposed by  leveraging the window-wise attention to enhance the learning and generation of time series. Experiments demonstrate performance improvement on 12 real-world .datasets

**Strengths:**

1. well-structured, clear presentation and easy-to-follow.

2. impressive experimental results.

**Weaknesses:**

1. According to Figure 1, the difference between point-wise, patch-wise, and window-wise operations can be regarded as convolutions with different kernel sizes and strides. Specifically, setting the stride smaller than the kernel size could also model "middle-level interactions". Also, the patch operation has complexity advantages (as it needs fewer tokens for attention calculation). This needs more discussion and analysis.

2. Why choose Maximum Mean Discrepancy (MMD) and Kullback-Leibler Divergence (K-L) for experimental evaluation? How about other metrics?

3. According to Figure 4, it seems TimeD better captures some high-frequency patterns, while Winformer captures the overall coarse-grained temporal variations.

**Questions:**

please refer to the weakness.

---

> ### Author Response · Authors · 2025-11-20
> **Official Comment by Authors**
>
> Dear reviewer z9c8:
>
> Thank you for this insightful questions. We are really appreciate that your input could enrich our analysis.
>
> ---
>
> ### W1: Discussion on three kinds of operations
>
> > W1: According to Figure 1, the difference between point-wise, patch-wise, and window-wise operations can be regarded as convolutions with different kernel sizes and strides. Specifically, setting the stride smaller than the kernel size could also model "middle-level interactions". Also, the patch operation has complexity advantages (as it needs fewer tokens for attention calculation). This needs more discussion and analysis.
>
> **W1.1: Differences of three operations**
>
>  We agree the three operations share convolution-like traits, but key differences lie in **alignment logic**: patch-wise uses fixed, non-overlapping kernels (stride=kernel size) that fragment cycles (Fig.1(b)), while window-wise adopts sliding kernels (stride<kernel size) with frequency-domain alignment (Ample attention) to preserve continuity (Fig.1(c)). The window-wise form is more unified, while the **patch-wise and point-wise forms are special cases of the window-wise alignment**. These special cases perform well under specific circumstances, but in cross-domain tasks, due to differences in the periodic patterns of different domains, a simple patch-wise approach cannot accommodate the cycles of all domains. In contrast, our method is more flexible, so **our approach allows for more flexible collection of cycles without the need for elaborate design**.
>
> **W1.2: Hyper-parameter study for "middle-level interactions"**
>
> To further explore the setting of "middle-level interactions", where the stride smaller than the kernel size, we evaluated the performance with different strides. The results are shown in the following table. Due to time constraints, these methods were only tested on a few runs with partial seeds with MMD metric. We can find that, stride=1 is optimal, achieving 7 best performance among 12 datasets. The reason is that, **with larger stride, partial features are lost between two kernels windows**. For example, when we set stride=2, we can take the structure in [1,2,3] and [3,4,5], but the structure of [2,3,4] is lost. These middle-level interactions may show effectiveness in some specific domains. Consider the cross-domain behavious, set stride=1 is optimal.
>
> **Table 4.1 Hyper-parameter study for window strides.**
>
> | **dataset** | **s=1**      | **s=2**      | **s=3**  | **s=4**      | **s=6**  | **s=12**     | **s=24**     |
> | ----------- | ---------- | ---------- | ------ | ---------- | ------ | ---------- | ---------- |
> | electricity | 0.0077     | **0.0068** | 0.0070 | 0.0071     | 0.0071 | 0.0069     | 0.0071     |
> | solar       | 0.0261     | 0.0246     | 0.0248 | 0.0250     | 0.0244 | **0.0243** | 0.0250     |
> | wind        | 0.0367     | 0.0349     | 0.0352 | 0.0350     | 0.0353 | **0.0347** | **0.0347** |
> | traffic     | **0.0710** | 0.0757     | 0.0804 | 0.0722     | 0.0844 | 0.0779     | 0.0745     |
> | taxi        | **0.0729** | 0.0801     | 0.0855 | 0.0827     | 0.0809 | 0.0873     | 0.0859     |
> | pedestrian  | **0.0422** | 0.0483     | 0.0481 | 0.0511     | 0.0487 | 0.0477     | 0.0485     |
> | air         | **0.0108** | 0.0124     | 0.0116 | 0.0116     | 0.0119 | 0.0113     | 0.0117     |
> | temperature | 0.2084     | 0.2080     | 0.2074 | 0.2075     | 0.2074 | **0.2061** | 0.2077     |
> | rain        | 0.0706     | 0.0284     | 0.0370 | **0.0244** | 0.0313 | 0.0325     | 0.0479     |
> | nn5         | **0.1451** | 0.1477     | 0.1485 | 0.1492     | 0.1464 | 0.1455     | 0.1466     |
> | fred_md     | **0.0016** | 0.0018     | 0.0019 | 0.0018     | 0.0018 | 0.0019     | 0.0018     |
> | exchange    | **0.1515** | 0.1563     | 0.1559 | 0.1552     | 0.1550 | 0.1559     | 0.1565     |
> | Best Count  | **7**      | 1          | 0      | 1          | 0      | 3          | 1          |
>
> **W1.3: About complexity**
>
> Patching could bring complexity advantages, so it's widely used in various time-seires modeling method. However, we think it squeeze the information when it reduce tokens, which means only $\frac{N}{p}$ tokens are reserved and the $i$-th token implies $X _{[(i-1)\*size+1, i\*size]}$. While our model takes sliding window, where $N$ tokens are reserved and the i-th token implies $X _{[i, i+size - 1]}$. **A reduction in computational complexity implies information compression or loss**. For patching methods, the segmentation and loss of periodic information is just caused by its reduction. Our method, with more computational consumption, could retain more compressive information.
>
> By the way, we further discussed the computaional complexity in Appendix B. The results shows window-wise alignment's consumption is acceptable (slight speed reduction compared to point-wise) due to framework (e.g. PyTorch) optimizations.

---

> ### Author Response · Authors · 2025-11-20
> **Official Comment by Authors**
>
> **W2 Metrics**
>
> > Why choose Maximum Mean Discrepancy (MMD) and Kullback-Leibler Divergence (K-L) for experimental evaluation? How about other metrics?
>
> **W2.1: Why MMD & KL**
>
> Choosing MMD and K-L for evaluation is due to their complementary strengths in measuring distribution similarity, which aligns with the goal of assessing time-series generation quality:
>
> **MMD**: As a non-parametric method, it avoids assuming specific distribution forms and leverages kernel functions to capture high-dimensional, non-linear distribution differences . It is well-suited for cross-domain scenarios, as it effectively quantifies discrepancies between real and generated time-series without requiring density estimation .
>
> **K-L Divergence**: It directly measures the information loss between two probability distributions, intuitively reflecting how much the generated data's distribution deviates from the real one . For time-series with interpretable periodic patterns, K-L helps reveal fine-grained differences in distribution shape.
>
> Together, they cover both flexibility (MMD) and interpretability (K-L), ensuring comprehensive evaluation of generation quality across diverse datasets .
>
> **W2.2: Other Metrics**
>
> In addition to these two distribution similarity measurements, we also incorporate the following two types of measuements: (1) Domain discrepancy: Measuring the diversity in the distribution of generated data across different domains, which is only for cross-domain generation. (2) Downstream task performance: Measuring whether the generated data can improve downstream tasks, i.e., whether using the generated data for data augmentation can enhance the model performance on prediction tasks.
>
> 1. Domain Discrepancy Measurement. We add a supplementary evaluation on domain discrepancy. Specifically, we evaluate the KL-divergence between different domains. The results are shown in Fig. 12 (added in Appendix D). To facilitate observation, all data in the figure are logarithmically transformed (log-transformed). For most of the datasets, the KL divergence between the generated data and the real data in their respective domains is the lowest, indicating that **our model can distinguish these domains and the domain-specific features are well learned**.
>
> 2. Downstream Task Performance. We have made a simple attempt to use the generated data for augmentation in forecasting. Due to time constraints, we only selected partial of the dataset with strong periodicity for testing. In our main results, we selected the length of 168 (24*7=168) as the total length for series generation. For downstream tasks, we select to kinds of settings: (1) using 24 * 6 = 144 points as historical observations, 24 points for prediction. (2) using  24 * 4 = 96 points as historical observations, 24 * 3=72 points for prediction. The pipeline is based on a popular time-series forecasting model, namely DLinear. The results are shown in following table, and the generated data augmentation **bring a improvement, with 29.62% in MAE**. This indicates that our model provides certain assistance for downstream tasks and can facilitate the training of foundational models or task-specific models under the condition of insufficient data.
>
>
> **Table 4.2 Results of time-series forecasting with/without augmented data generated by our winformer.**
>    | dataset     | length | Augmented(Ours)|  | Non-Augmented |  |
>    | ----------- | ------ | --------------------- | --------------------- | ----------- | ----------- |
>    |||MSE|MAE|MSE|MAE
>    | electricity | 24     | **0.008**                 | **0.017**                 | 0.009       | 0.019       |
>    | | 72     | 0.010                 | 0.019                 | **0.009**       | **0.018**       |
>    | exchange    | 24     | **0.007**                 | **0.054**                 | 0.138       | 0.281       |
>    | | 72     | **0.006**                 | **0.049**                 | 0.071       | 0.204       |
>    | taxi        | 24     | 0.171                 | 0.275                 | **0.169**       | **0.282**       |
>    | | 72     | **0.316**                 | **0.370**                 | 0.333       | 0.384       |
>    | traffic     | 24     | **0.494**                 | **0.461**                 | 0.637       | 0.527       |
>    | | 72     | **0.517**                | **0.395**                 | 0.595       | 0.434       |

---

> ### Author Response · Authors · 2025-11-20
> **Official Comment by Authors**
>
> **W3: Discussion of TimeDP and Winformer**
>
> > According to Figure 4, it seems TimeD better captures some high-frequency patterns, while Winformer captures the overall coarse-grained temporal variations.
>
> Thank you for your observation.
>
> 1. Though TimeDP captures high-frequency patterns, it **may not beneficial for cross-domain generation**. As we know, forecasting (or imputation) tasks focus on estimating the series to the groundtruth under know observations in every single instances, which need detailed perception. While generation is to **estimate the global distribution** to the whole dataset, which imposes **stronger requirements on more widely existing structures awareness**. Thus, some global features for distribution are more important, because the generation model targets on estimate the generated distribution to the real distribution. As winformer could capture these global features, it achieves better performance with 10.67% average MMD gain over TimeDP.
>
> 2. High-frequency patterns are often noise. For some domains, **high-frequency patterns are often contaminated  with domain-specific noise diluting the meaningful signals**. Some researches have mentioned this[1][2], and it's challenging to extract meaningful high-frequency signals from the high-frequency noise. TimeDP may may only mechanically retain these high-frequency signals, thus introducing noise, which could be another reason why our Winformer show better results.
>
> As analyzed above, we explained why there is no improvement for TimeDP to replicates high-frequency signals. And we think it's a good view to compare the methods and understand the model from another perspective.
>
> [1] Kun Yi, Jingru Fei, Qi Zhang, Hui He, Shufeng Hao, Defu Lian, Wei Fan. FilterNet: Harnessing Frequency Filters for Time Series Forecasting. NeurIPS 2024.
> [2] Kun Yi, Qi Zhang, Wei Fan, Shoujin Wang, Pengyang Wang, Hui He, Ning An, Defu Lian, Longbing Cao, Zhendong Niu. Frequency-domain MLPs are More Effective Learners in Time Series Forecasting. NeurIPS 2023.

---

> > ### Comment · Reviewer_z9c8 · 2025-11-26
> >
> > Thanks to the author for the detailed response. I'll keep my positive rating.

---

### Official Review · Reviewer_6vwQ · 2025-10-29

**Soundness:** 3
**Presentation:** 3
**Contribution:** 3
**Rating:** 6
**Confidence:** 3

**Summary:**

This paper introduces Winformer, a diffusion-based Transformer for cross-domain time-series generation. It proposes the Ample Attention, which extends point-wise similarity to window-wise comparison in the frequency domain, enabling better modeling of periodic and long-range dependencies. Experiments on 12 datasets show consistent improvements, with an average 10.67% gain over strong baselines.

**Strengths:**

- The proposed attention mechanism is conceptually novel and addresses a clear limitation of pairwise similarity in Transformers.

- The integration of window-wise attention into a diffusion framework is technically sound and well motivated.

- Strong empirical results and ablation studies demonstrate robustness and broad applicability across domains.

**Weaknesses:**

The paper focuses exclusively on evaluating generative quality (MMD, visual consistency) but lacks experiments on downstream utility. Demonstrating whether the generated data can enhance forecasting or imputation would significantly strengthen the practical impact of the proposed model.

**Questions:**

See weaknesses.

---

> ### Author Response · Authors · 2025-11-20
> **Official Comment by Authors**
>
> Dear reviewer  6vwQ:
>
> We highly appreciate your recognition of our paper. We will make every effort to address your questions and refine the content of the paper.
>
> ---
>
> ### Downstream tasks
>
> > The paper focuses exclusively on evaluating generative quality (MMD, visual consistency) but lacks experiments on downstream utility. Demonstrating whether the generated data can enhance forecasting or imputation would significantly strengthen the practical impact of the proposed model.
>
> We have made a simple attempt to use the generated data for augmentation in forecasting. Due to time constraints, we only selected partial of the dataset with strong periodicity for testing. In our generation experiments, we selected the length of 168 (24*7=168) as the total length for series generation. For downstream tasks, we select two kinds of settings: (1) using 24 * 6 = 144 points as historical observations, 24 points for prediction. (2) using  24 * 4 = 96 points as historical observations, 24 * 3=72 points for prediction. We conduct the experiments based on the pipeline of DLinear. The results are shown in following table, and the generated data augmentation **bring a improvement, with averagely 29.62% in MAE** across 4 datasets. **This indicates that our model provides certain assistance for downstream tasks and can facilitate the training of foundational models or task-specific models under the condition of insufficient data.**
>
> **Table 3.1 Results of time-series forecasting with/without augmented data generated by our winformer.**
>
>    | dataset     | length | Augmented(Ours)|  | Non-Augmented |  |
>    | ----------- | ------ | --------------------- | --------------------- | ----------- | ----------- |
>    |||MSE|MAE|MSE|MAE
>    | electricity | 24     | **0.008**                 | **0.017**                 | 0.009       | 0.019       |
>    | | 72     | 0.010                 | 0.019                 | **0.009**       | **0.018**       |
>    | exchange    | 24     | **0.007**                 | **0.054**                 | 0.138       | 0.281       |
>    | | 72     | **0.006**                 | **0.049**                 | 0.071       | 0.204       |
>    | taxi        | 24     | 0.171                 | 0.275                 | **0.169**       | **0.282**       |
>    | | 72     | **0.316**                 | **0.370**                 | 0.333       | 0.384       |
>    | traffic     | 24     | **0.494**                 | **0.461**                 | 0.637       | 0.527       |
>    | | 72     | **0.517**                | **0.395**                 | 0.595       | 0.434       |

---

### Official Review · Reviewer_tGxi · 2025-10-31

**Soundness:** 3
**Presentation:** 3
**Contribution:** 2
**Rating:** 4
**Confidence:** 3

**Summary:**

This paper proposes Winformer, a diffusion-based model for time-series generation, particularly in cross-domain settings. It introduces "Ample attention," a window-wise attention mechanism that extends traditional pairwise similarity computations to comparisons between sliding windows of time-series data, leveraging Fourier transforms to capture periodic patterns and mitigate time-warping issues. The framework is built on a Transformer architecture tailored for denoising in diffusion probabilistic models. The authors claim this approach better handles complex trends, periods, and noise across domains compared to point-wise or patch-wise methods.

**Strengths:**

1. The window-wise attention is an interesting extension of standard self-attention, potentially better suited for time-series data with inherent periodicity and warping, as demonstrated through Fourier-based derivations.

2. The focus on cross-domain generation addresses a challenging problem relevant to applications like data imputation and domain adaptation.

3. Theoretical insights (e.g., derivations in Section 3.1) provide some grounding for the proposed Ample attention, with promises of further proofs in the appendix.

**Weaknesses:**

1. Limited Scope of Applicability: While the paper motivates time-series generation as a foundational step for broader applications (e.g., imputation, feature augmentation, domain adaptation, and foundation modeling), the proposed methodology appears tightly coupled to diffusion-based generation tasks, particularly imputation via denoising. The Ample attention and window-wise processing are innovative for capturing periodic patterns in generative settings, but the paper does not demonstrate or discuss their utility in other core time-series tasks, such as forecasting, anomaly detection, classification, or clustering. For instance, it's unclear how Winformer would integrate into non-diffusion pipelines, where attention mechanisms typically operate differently. This narrow evaluation raises questions about the generalizability of the contributions to the wider time-series field, potentially limiting its impact beyond specialized generation scenarios.

2. Methodological Novelty and Integration: The Ample attention builds on established ideas like Fourier decompositions (e.g., from Alaa et al., 2021) and window-based processing, but the expansion to window-to-window alignments via learnable convolutions feels somewhat incremental rather than transformative. The overall architecture (e.g., combining diffusion with Transformer layers) echoes recent works like CSDI or TimeDiff, and it's not evident how the window-wise shift uniquely resolves limitations in pairwise or patch-wise approaches beyond empirical gains. Additionally, hyperparameters like window size (p) and stride (s) could introduce sensitivity, but without thorough ablations, their robustness is unclear.

3. Experimental Validation: The claimed average improvement is promising, but the experiments seem confined to cross-domain generation metrics, without extensions to downstream tasks. .The 12 datasets are diverse, but details on domain shifts or failure cases are limited in the provided sections. Cross-domain focus is emphasized, but without metrics like domain discrepancy measures.

4. The writing is clear but could better visualize the attention expansion process beyond Figures 1-2.

**Questions:**

1. Beyond diffusion-based imputation, how could Ample attention be adapted for other time-series tasks like forecasting or classification? Have you experimented with integrating it into non-generative models?

2. How sensitive is the model to window size (p) and stride (s)? Ablations on these would help.

---

> ### Author Response · Authors · 2025-11-20
> **Official Comment by Authors**
>
> Dear reviewer tGxi:
>
> Thank you for your recognition of this paper's soundness and presentation. We will try our best to respond to your questions one by one.
>
> ---
>
> ### W1: Scope of Applicability (Also Q1)
>
> > W1: Limited Scope of Applicability: While the paper motivates time-series generation as a foundational step for broader applications (e.g., imputation, feature augmentation, domain adaptation, and foundation modeling), the proposed methodology appears tightly coupled to diffusion-based generation tasks, particularly imputation via denoising. The Ample attention and window-wise processing are innovative for capturing periodic patterns in generative settings, but the paper does not demonstrate or discuss their utility in other core time-series tasks, such as forecasting, anomaly detection, classification, or clustering. For instance, it's unclear how Winformer would integrate into non-diffusion pipelines, where attention mechanisms typically operate differently. This narrow evaluation raises questions about the generalizability of the contributions to the wider time-series field, potentially limiting its impact beyond specialized generation scenarios.
>
> Thank you for your question. To explore the applicability, we conduct the following experiments. **Integrating Ample attention into forecasting pipeline:** We have made a simple attempt to integreate our Ample attention into non-generation pipelines. We transplanted ample attention to long-term series forecasting tasks and compared it with the vanilla attention to verify whether it can bring certain performance improvements. As results shown in the following Table, it can be observed that our method bring an improvement on 5 datasets, with averagely 5.2% improvement on MSE and 2.9% improvement on MAE. **Without task-specific designing, simply adapting the Ample attention can already get a certain performance improvement**. We believe that with further elaborate design, this method is promising to bring more significant enhancements to downstream tasks.
>
> **Table 2.1 Results of integrating our Ample attention into forecasting pipelines**
>
> | dataset | length | Ample Attn |  | Vanilla Attn |   |
>    | ------- | ------ | ------------------------ | ------------------------ | ----------- | ----------- |
>    |         |        | MSE                      | MAE                      | MSE         | MAE         |
>    | ETTh1   | 336    | **0.3360**                   | **0.5099**                   | 0.4113      | 0.5732      |
>    |    | 720    | **0.3200**                   | **0.5018**                   | 0.3212      | 0.5031      |
>    | ETTh2   | 336    | **0.2173**                   | **0.3778**                   | 0.2204      | 0.3804      |
>    |    | 720    | **0.2466**                   | **0.4061**                   | 0.2474      | 0.4067      |
>    | ETTm1   | 336    | 0.2336                   | 0.4159                   | **0.2298**      | **0.4117**      |
>    |    | 720    | **0.2054**                   | **0.3793**                   | 0.2126      | 0.3874      |
>    | ETTm2   | 336    | **0.1461**                   | **0.3007**                   | 0.1480      | 0.3020      |
>    |    | 720    | **0.1898**                   | **0.3505**                   | 0.2139      | 0.3693      |
>    | ECL     | 336    | **0.4236**                   | **0.4785**                   | 0.4784      | 0.5135      |
>    |      | 720    | **0.4522**                   | **0.5088**                   | 0.4803      | 0.5279      |
>
> Moreover, we want to note that our model is not a foundational model for time-series. Generation is a foundational step for broader applications, but it may still take efforts to apply generative models to downstream tasks as there are task-specific charateristics.  Therefore, our model currently focuses more on the field of cross-domain generation, and we would appreciate your understanding. And in future work, we will continue to investigate how to enable this set of models to better adapt to downstream tasks.

---

> ### Author Response · Authors · 2025-11-20
> **Official Comment by Authors**
>
> ### W2: About Novelty and Integration (1/2)
>
> Since this question involves multiple sub-questions, we will respond to each sub-question one by one in 2 posts. This is the 1st post。
>
> > W2: Methodological Novelty and Integration: The Ample attention builds on established ideas like Fourier decompositions (e.g., from Alaa et al., 2021) and window-based processing, but the expansion to window-to-window alignments via learnable convolutions feels somewhat incremental rather than transformative. The overall architecture (e.g., combining diffusion with Transformer layers) echoes recent works like CSDI or TimeDiff, and it's not evident how the window-wise shift uniquely resolves limitations in pairwise or patch-wise approaches beyond empirical gains. Additionally, hyperparameters like window size (p) and stride (s) could introduce sensitivity, but without thorough ablations, their robustness is unclear.
>
>
> **W2.1 About Novelty**
>
> Winformer's novelty lies in fusing frequency-based similarity with learnable convolutions for adaptive window alignment. Although the final changes **is simple, but it's effective** because it is derived from careful observations and theoretical deduction. Thus, we would like to further discuss the motivation and theoretical derivation of it.
>
> **Motivation**: As illustraed in Fig. 13 (added in Appendix F), we try to explain how we get the observation and the motivation. The design of Winformer's adaptive window-wise architecture stems from a key observation:  Traditional point-wise attention struggles with complex dependencies and fails to model cyclic structures when time-series are numerically sensitive in continues space. Patch-wise methods fragment time-series into fixed segments, breaking evolving periodic patterns because patching with a fixed segments could not be suitable for every domain. These flaws motivated us to rethink the "processing unit" of attention: Could we shift the discrete points or patches to continuous windows that naturally encapsulate periodic cycles? Compared to the point-wise methods, the window-wise methods can summary local features and simplified attention's work. Compared to the patch-wise methods, the window alignment is sliding, which allows to adapt different domains, as for every domain there is  expected to have partial of the windows can contain the proper periodicity features.
>
> **Theoretical Derivation**: With these observations, we start for theoretical analysis. Time-series periodicity is inherently a frequency-domain property, so we leveraged the frequency transform to decompose sequences into spectral components, enabling similarity comparison across windows. We decompose the calculation of the **frequency transform operator** as a linear transformation $\mathcal{F} _{D} ( \mathbf{x} ) = \mathbf{M} \mathbf{x}$, where $M$ is the matrix derived from Fourier transform basis. Then, we consider to adopt it on attention maps by $ \widetilde{\mathbf{F}} _{(t_1,t_2)}^{(p,q)} = \mathcal{F} _{D} \lbrace [\mathbf{Q}] _{t_1}^{(q)} \rbrace \cdot \mathcal{F} _{D} \lbrace [\mathbf{K}] _{t_2}^{(q)} \rbrace$. And we get the real part as the **similarity for attention** as $\mathbf{S} _{(t_1,t_2)}$. After a series of derivations, we calculate $\mathbf{S} _{t_1,t_2}$ as $
> \mathbf{S} _{(t_1,t_2)} =  \frac{\mathbf{W}^{\top}}{\sqrt{p}}  \sum [  \textrm{avg} (\mathbf{S}^{'}) + \textrm{conv} _{\psi(1)} (\mathbf{S}^{'}) +\cdots +  \textrm{conv} _{\psi(p-1)} (\mathbf{S}^{'}) ]$, and $conv _{\psi(i)}$($\cdot$) represents the convolution operator with the kernel $\psi(i)$. Thus, we can acquire the window-wise score by performing the convolution operator on the original attention score.
>
> **W2.2 More Evidences beyond empirical gains.**
>
> Beyond empirical gains, we would like to discuss it from the visualization of intermediate processes and visualization of the generated samples.
>
> 1. **Visualization of intermediate processes**: As illustrated in Fig 8 (in Appendix C.1.2), there are attention maps before and after our special initialized convolutional layer. We can find that, our frequancy-aware convolutional layer **enhance the periodicity** by showing larger grids. Without our approach, attention just change the organization form of the data input, which could not achieve such outcomes.
>
> 2. **Visualization of the generated samples among point-wise, path-wise and window-wise alignment**: We add the visualized comparisons of generated series across the three kinds alignments in Appendix C.2. We conducted visualization on taxi dataset, which contains complex periodicity structure. In this datasets, series not only present the daily cycles but also includes the small periodic changes of morning and evening peaks. We visualized the series generated with point-wise, patch-wise and window-wise alignment in Fig.9 (added in Appendix C.2). We can find that, window-wise method could **capture the periodicity better than other methods, by well establishing periodicity structures**.

---

> ### Author Response · Authors · 2025-11-20
> **Official Comment by Authors**
>
> ### W2: About Novelty and Integration (2/2)
>
> Since this question involves multiple sub-questions, we will respond to each sub-question one by one in 2 posts. This is the 2nd post。
>
> > W2: Methodological Novelty and Integration: The Ample attention builds on established ideas like Fourier decompositions (e.g., from Alaa et al., 2021) and window-based processing, but the expansion to window-to-window alignments via learnable convolutions feels somewhat incremental rather than transformative. The overall architecture (e.g., combining diffusion with Transformer layers) echoes recent works like CSDI or TimeDiff, and it's not evident how the window-wise shift uniquely resolves limitations in pairwise or patch-wise approaches beyond empirical gains. Additionally, hyperparameters like window size (p) and stride (s) could introduce sensitivity, but without thorough ablations, their robustness is unclear.
>
> **W2.3 Hyper-parameter study (Also Q2)**
>
> We evaluated the performance with different kernel sizes and strides. The results are shown in following tables with MMD metrics.
>
> 1. **For window size**: We add the hyper-parameter study about the window size in Appendix A, following table is partial of the results with MMD metric in Table 3 (Appendix A). Firstly, we can find that DCT kernel (15 best) performs better than Avg pooling (11 best), because DCT kernel could capture more periodicity information. Secondly, the kernel size with 25 for DCT is recommended, as it could cover common cycles of these datasets. The optimal 25-size kernel in the paper works because it includes typical cycles (4, 6, 12, 24) without overcomplicating computation.
>
> **Table 2.2 Hyper parameter study for window sizes.**
>
> |             | conv,p=7 | conv,p=13 | conv,p=25 | avg,p=7 | avg,p=13 | avg,p=25 |
> | ----------- | --------- | ---------- | ---------- | -------- | --------- | --------- |
> | Electricity | **0.001**     | 0.002 | **0.001**| 0.002    | 0.002     | 0.002  |
> | Solar       | 0.035     | 0.035      | 0.035      | **0.033**    | **0.033**     | **0.033**     |
> | Wind        | 0.035     | **0.033**      | 0.034      | **0.033**   | **0.033**     | 0.037     |
> | Traffic     | 0.074     | 0.074      | **0.071**      | 0.072    | **0.071**     | 0.074     |
> | Taxi        | **0.080**     | 0.086      | 0.085      | 0.084    | 0.083     | 0.082     |
> | Pedestrain  | **0.040**     | 0.041      | **0.040**      | 0.042    | 0.041     | **0.040**     |
> | Air         | 0.014     | 0.013      | **0.011**      | 0.013    | 0.013     | 0.012     |
> | Temperature | 0.226     | **0.219**      | 0.230      | 0.220    | **0.219**     | 0.229     |
> | Rain        | **0.033**     | 0.050 | 0.036 | 0.038 | 0.037 | 0.036     |
> | NN5         | 0.149     | 0.151| **0.147** | 0.153 | 0.154  | 0.154     |
> | Fred-MD     | **0.002**| **0.002** | **0.002** | **0.002**    | **0.002**     | **0.002**     |
> | Exchange    | **0.136**| 0.139  | 0.137| 0.140    | 0.139     | 0.138     |
>
> 2. **For stride**: We report the hyper-parameter about the stride as following. Due to time constraints, these methods were only tested with a few runs with partial seeds with MMD. We can find that, stride=1 is optimal, achieving 7 best performance among 12 datasets. The reason is that, with more stride, partial features are lost between two kernels windows. Thus, when we set stride=1, the model get  the most comprehensive information, and the model performs better than others.
>
> **Table 2.3 Hyper-parameter study for window strides.**
>
> | **dataset** | **s=1**      | **s=2**      | **s=3**  | **s=4**      | **s=6**  | **s=12**     | **s=24**     |
> | ----------- | ---------- | ---------- | ------ | ---------- | ------ | ---------- | ---------- |
> | electricity | 0.0077     | **0.0068** | 0.0070 | 0.0071     | 0.0071 | 0.0069     | 0.0071     |
> | solar       | 0.0261     | 0.0246 | 0.0248 | 0.0250  | 0.0244 | **0.0243** | 0.0250     |
> | wind        | 0.0367     | 0.0349 | 0.0352 | 0.0350  | 0.0353 | **0.0347** | **0.0347** |
> | traffic     | **0.0710** | 0.0757  | 0.0804 | 0.0722 | 0.0844 | 0.0779     | 0.0745     |
> | taxi        | **0.0729** | 0.0801     | 0.0855 | 0.0827| 0.0809 | 0.0873     | 0.0859     |
> | pedestrian  | **0.0422** | 0.0483     | 0.0481 | 0.0511     | 0.0487 | 0.0477     | 0.0485     |
> | air         | **0.0108** | 0.0124     | 0.0116 | 0.0116     | 0.0119 | 0.0113     | 0.0117     |
> | temperature | 0.2084     | 0.2080     | 0.2074 | 0.2075     | 0.2074 | **0.2061** | 0.2077     |
> | rain        | 0.0706     | 0.0284     | 0.0370 | **0.0244** | 0.0313 | 0.0325     | 0.0479     |
> | nn5         | **0.1451** | 0.1477     | 0.1485 | 0.1492     | 0.1464 | 0.1455     | 0.1466     |
> | fred_md     | **0.0016** | 0.0018     | 0.0019 | 0.0018     | 0.0018 | 0.0019     | 0.0018     |
> | exchange    | **0.1515** | 0.1563     | 0.1559 | 0.1552     | 0.1550 | 0.1559     | 0.1565     |
> | Best Count  | **7**      | 1          | 0      | 1          | 0      | 3          | 1          |

---

> ### Author Response · Authors · 2025-11-20
> **Official Comment by Authors**
>
> ### W3: Experimental Validation
>
> > W3: Experimental Validation: The claimed average improvement is promising, but the experiments seem confined to cross-domain generation metrics, without extensions to downstream tasks. .The 12 datasets are diverse, but details on domain shifts or failure cases are limited in the provided sections. Cross-domain focus is emphasized, but without metrics like domain discrepancy measures.
>
> **W3.1 Evaluation on Downstream tasks**
>
> We have made an attempt to use the generated data for augmentation in forecasting task. Due to time constraints, we only selected partial of the dataset with strong periodicity for testing. In our main results, we selected the length of 168 (24*7=168) as the total length for series generation. For downstream tasks, we select two kinds of settings: (1) using 24 * 6 = 144 points as historical observations, 24 points for prediction. (2) using  24 * 4 = 96 points as historical observations, 24 * 3=72 points for prediction. The pipeline is based on DLinear[1], which is a popular time-series forecasting model. The results are shown in following table, and the generated data augmentation **bring an improvement, with 29.62% in MAE**. This indicates that our model provides certain assistance for downstream tasks and can facilitate the training of foundational models or task-specific models under the condition of insufficient data.
>
> **Table 2.4 Results of time-series forecasting with/without augmented data generated by our winformer.**
>
>    | dataset     | length | Augmented(Ours)|  | Non-Augmented |  |
>    | ----------- | ------ | --------------------- | --------------------- | ----------- | ----------- |
>    |||MSE|MAE|MSE|MAE
>    | electricity | 24     | **0.008**                 | **0.017**                 | 0.009       | 0.019       |
>    | | 72     | 0.010                 | 0.019                 | **0.009**       | **0.018**       |
>    | exchange    | 24     | **0.007**                 | **0.054**                 | 0.138       | 0.281       |
>    | | 72     | **0.006**                 | **0.049**                 | 0.071       | 0.204       |
>    | taxi        | 24     | 0.171                 | 0.275                 | **0.169**       | **0.282**       |
>    | | 72     | **0.316**                 | **0.370**                 | 0.333       | 0.384       |
>    | traffic     | 24     | **0.494**                 | **0.461**                 | 0.637       | 0.527       |
>    | | 72     | **0.517**                | **0.395**                 | 0.595       | 0.434       |
>
> **W3.2 Details on domain shifts or failure case**
>
> We conducted the time-series generation experiment on 12 real-world datasets from four domains, including energy, economic, weather and transportation. The details of the datasets are reported in Appendix E.2.1. The time-series from these datasets **show differences in periodicity**. For example, the solar dataset is sampled daily, with macro cycle of 24. Since the intensity of solar energy varies with the solar cycle and exhibits strong regularity, its periodic structure is simple. However, it's quite different for taxi dataset. Taxi dataset is sampled every 30min, thus it reveals a macro cycle of 48. Moreover, due to morning/evening peaks and society factors, there more detailed structure changes and micro cycles, leading to complex structure. In summary, due to domain characteristic, these time series **contain different sizes of the cycles with complexity of cycle superposition**.
>
> We also visualized the failure cases that existing point-wise or patch-wise methods may fail in capturing the complex periodicity. We have discussed about the failure cases and how our method resolve these limitations in W2.2 (2). We visualized the series generated with point-wise, patch-wise and window-wise alignment in Fig.9 (added in Appendix C.2). We can find that, window-wise method could capture the periodicity better than other methods, by well establishing periodicity structures.
>
> **W3.3 Domain Discrepancy Measures**
>
> We add a supplementary evaluation on domain discrepancy. Specifically, we evaluate the KL-divergence between different domains. The results are shown in Fig. 12 (added in Appendix D). To facilitate observation, all data in the figure are logarithmically transformed (log-transformed). For most of the datasets, the KL divergence between the generated data and the real data in their respective domains is the lowest, indicating that our model can **distinguish these domains** and the **domain-specific features are well learned**.
>
> [1] Ailing Zeng, Muxi Chen, Lei Zhang, Qiang Xu. Are Transformers Effective for Time Series Forecasting? AAAI 2023: 11121-11128

---

> ### Author Response · Authors · 2025-11-20
> **Official Comment by Authors**
>
> **W4: Better visualization**
>
> > W4: The writing is clear but could better visualize the attention expansion process beyond Figures 1-2.
>
> Thank you for your suggestions. We refine the diagrams to achieve a clearer presentation.
> 1. We have added more detailed explanations in Figure 3, including how to use convolutional layers to perform window-wise alignment on the attention map.
> 2. We also visualized the attention maps in Fig.6 (added in Section 5), the sub figures show how the attention map changes during training process.
> 3. We add the differences of point-wise and patch-wise methods in the Figure 13 (added in Appendix F) to further explain the motivation of the Ample attention.
> 4. We updated some visualization results as Figure 9 and 12, which may help for more clearer analysis.
>
> **Q1: Adaptation on downstream tasks**
>
> > Beyond diffusion-based imputation, how could Ample attention be adapted for other time-series tasks like forecasting or classification? Have you experimented with integrating it into non-generative models?
>
> Thank you for your question, we find it's the **same with W1**, and we have discussed the adaptation results for downstream tasks on this issue in W1. We find that our Ample attention could benefit the forecasting model even without any task-specific designation. More detailed results could be found in W1.
>
> **Q2: Hyper-parameter study for window size and stride**
>
> > How sensitive is the model to window size (p) and stride (s)? Ablations on these would help.
>
> Thank you for your question, we find it's the **same with W2.3**, and we have shown the hyper-parameter study and the related analysis on this issue in W2.3. Based on the hyper-parameter analysis, we choose to use the optimal setting with dct kernel, p=25, s=1. More detailed results and analysis could be found in W2.3

---

### Official Review · Reviewer_gSxW · 2025-10-31

**Soundness:** 3
**Presentation:** 1
**Contribution:** 3
**Rating:** 4
**Confidence:** 4

**Summary:**

This paper aims to address the problem in existing cross-domain generation method that failing to capture and adapt to complex periodic patterns of diverse domains. This paper presents Winformer, which calculates attention scores between windows instead of between individual points, to enhance perception of complex time series patterns. The proposed window-wise attention mechanism, called Ample attention, is introduced as calculating similarity score between the Discrete Fourier Transform results of each window and is implemented by convolution on original attention scores. Experiments are conducted on 12 real-world time series datasets, in comparison with six time series generation baselines on two metrics. The ablation studies show the advantages of the proposed mechanism and discussions show how periodicity is captured and utilized by Winformer.

**Strengths:**

1. This paper targets on improving of attention mechanism of time series modeling, which is not only meaningful for cross-domain time series generation task, but also potentially beneficial for time series forecasting or other applications.
2. The ample attention is reasonably designed, providing a new option for time series transformer that emphasizes window-by-window correlations.
3. The experiments are extensive and comprehensive, covering multiple datasets and baselines. The discussions also help understand the advantage of the proposed Winformer method.

**Weaknesses:**

1. Many sentences in this paper are incoherent and difficult to follow. It feels like it hasn't been proofread for clarity. Some syntactic refinement would greatly improve it. There is also a structuring problem, i.e. missing a conclusion sector.
2. The motivations for Winformer and some other claims are not well-supported. The authors state that they “find that a more adaptive architecture works better for coupling trending and periodic patterns”. But as a motivation, we would like to know about how is this architecture discovered and what is the theoretical intuition behind the design. Similarly, why cross-domain generation task is better for demonstrate the effectiveness, instead of regular/long-term time series forecasting task or time series imputation task, should be properly justified.
3. The convolution mechanism has introduced additional computation overhead, while the performance improvement against its point-wise attention variant is very significant. The paper can benefit from providing a more thorough analysis (e.g., visualized comparison of generated sequences) into the failure mode of window-wise, patch-wise and point-wise attention respectively, to show the advantage of Winformer more clearly, rather than just claiming “periodic feature in this dataset is weak” without justification.

**Questions:**

1. How do you describe the differences among point-wise, patch-wise and window-wise attention map, and how to explain these differences?
2. Are results sensitive to the convolution kernel initialization method? How do the kernel weights change along the training process?
3. As the results for different kernel sizes are close to each other, can the authors provide some general rules for selecting this hyper-parameter?

---

> ### Author Response · Authors · 2025-11-20
> **Official Comment by Authors**
>
> Dear Reviewer gSxW:
>
> We really appreciate your recognition of this paper's soundness and contributions. We try our best to response each of the issues one by one next.
>
> ----
>
> ### W1: syntactic refinement & Structure Problem
>
> > W1: Many sentences in this paper are incoherent and difficult to follow. It feels like it hasn't been proofread for clarity. Some syntactic refinement would greatly improve it. There is also a structuring problem, i.e. missing a conclusion sector.
>
> Thank you for your suggestions. We have refined the sentences for better flow (highlighted in blue in the main text and appendix) and supplemented the conclusion section as you recommended (In Section 7).
>
> ---
>
> ### W2: Motivations
>
> > W2: The motivations for Winformer and some other claims are not well-supported. The authors state that they “find that a more adaptive architecture works better for coupling trending and periodic patterns”. But as a motivation, we would like to know about how is this architecture discovered and what is the theoretical intuition behind the design. Similarly, why cross-domain generation task is better for demonstrate the effectiveness, instead of regular/long-term time series forecasting task or time series imputation task, should be properly justified.
>
> Thank you for your suggestions. Below, we will clarify the main flow of this paper regarding task selection, motivation derivation, and theoretical deduction. Additionally, we have revised the relevant sentences in the main text and appendix, hoping this will help.
>
> **W2.1: Why cross-domain generation?**
>
> 1. **Forecasting VS Generation**:  Forecasting (and imputation) tasks focus on estimating the series to the groundtruth under know observations in every single instances, which need detailed perception. While generation is to estimate the global distribution to the whole dataset, which imposes stronger requirements on more widely existing structures awareness. That's why we select the generation tasks to explore how could the model learn the periodicity structure better.
>
> 2. **Cross-domain generation**: Cross-domain generation requires models to learn and transfer periodic patterns across diverse domains. As illustrated in Fig. 13(a) (updated in Appendix F), series from domain 1 and domain 2 show different periodicity, which is challenging for existing point-wise and patch-wise methods. Capturing the domain-specific periodicity directly tests the architecture's ability to generalize periodicity.
>
> **W2.2: How is the architecture discovered?**
>
> The design of Winformer's window-wise architecture stems from a key observation: Traditional point-wise attention struggles with complex dependencies and fails to model cyclic structures. As illustraed in Fig. 13(b) (added in Appendix F), it's **difficult to identify repetitive structures when time-series are numerically sensitive in continues space**. Patch-wise methods fragment time-series into fixed segments, breaking periodic patterns . As illustrated in Fig. 13(c) (added in Appendix F), for cross-domain generation, time-series have different cycles, **patching with a fixed segments could not be suitable for every domain** . These flaws motivated us to **rethink the "processing unit" of attention**: Could we shift the discrete points or patches to continuous sliding windows that naturally encapsulate periodic cycles? Compared to the point-wise methods, the window-wise methods can summary local features and simplified attention's work. Compared to the patch-wise methods, the window alignment is sliding, which allows to adapt different domains, as for every domain there is expected to have partial of the windows can contain the proper periodicity features.
>
> **W2.3 What is the theoretical intuition behind the design**
>
> With these observations, we start for theoretical analysis. Time-series periodicity is inherently a frequency-domain property. We leveraged the frequency transform to decompose series into spectral components, enabling similarity comparison across windows. We decompose the calculation of the frequency transformation as a linear transformation:
> $ \mathcal{F} _{D}(\mathbf{x})=\mathbf{M} \mathbf{x} $, where $ M $ is the matrix derived from transform basis. Then, we consider to adopt it on attention maps by $ \widetilde{\mathbf{F}} _{(t_1,t_2)}^{(p,q)}=\mathcal{F} _{D}\lbrace[\mathbf{Q}] _{t_1}^{(q)}\rbrace{\cdot}\mathcal{F} _{D}\lbrace[\mathbf{K}] _{t_2}^{(q)}\rbrace $. We get the real part as the similarity for attention $ \mathbf{S} _{(t_1,t_2)} $. After a series of derivations, we calculate it as $ \mathbf{S} _{(t_1,t_2)}=\frac{\mathbf{W} ^{\top}}{\sqrt{p}}\sum[\textrm{avg}(\mathbf{S}^{'})+\textrm{conv} _{\psi(1)}(\mathbf{S}^{'})+\cdots+\textrm{conv} _{\psi(p-1)}(\mathbf{S}^{'})] $, where $ conv _{\psi(i)}(\cdot) $ represents the convolution operator with the kernel $ \psi(i) $. Thus, we can acquire the window-wise score by performing the convolution operator on the original attention score.

---

> ### Author Response · Authors · 2025-11-20
> **Official Comment by Authors**
>
> **W3: Comparison Analysis**
>
> > W3: The convolution mechanism has introduced additional computation overhead, while the performance improvement against its point-wise attention variant is very significant. The paper can benefit from providing a more thorough analysis (e.g., visualized comparison of generated sequences) into the failure mode of window-wise, patch-wise and point-wise attention respectively, to show the advantage of Winformer more clearly, rather than just claiming “periodic feature in this dataset is weak” without justification.
>
> Thank you for your suggestion. We fully agree that it would better highlight Winformer's advantages.
>
> We add the visualized comparisons of generated series across window-wise, patch-wise, and point-wise attention in Appendix C.2. To explore the differences in the effectiveness of the three kinds alignments, namely point-wise, patch-wise and window-wise, in extracting complex periodicity, we conducted result visualization on taxi dataset. Taxi dataset contains complex periodicity structure, as not only presents the daily cycle but also includes the detailed periodic changes of morning and evening peaks. We visualized the series generated with point-wise, patch-wise and window-wise alignment in Fig.9 (added in Appendix C.2). We can find that, window-wise method could capture the periodicity better than other methods, by well establishing clearer periodicity structures. Other methods show more chaotic periodicity.
>
> By the way, we have also supplemented specific computational cost in Appendix B, showing that the overhead of the convolution mechanism is acceptable, due to the optimizations of PyTorch for convolutional operators, which only a slight reduction in speed compared to point-wise attention.

---

> ### Author Response · Authors · 2025-11-20
> **Official Comment by Authors**
>
> **Q1: Differences among point-wise, patch-wise and window-wise attention map**
>
> > Q1: How do you describe the differences among point-wise, patch-wise and window-wise attention map, and how to explain these differences?
>
> The core differences for attention map lie in the size of "process unit" and how it moves:
>
> - **Point-wise (Fig. 1(a))**: (window size=1, stride=1). Attention map shows sparse point-to-point interactions. If we have two series $X_1$ and $X_2$ with length $N$, every 1 point construct one token, and the attention map is an $N \times N$  matrix.
> - **Patch-wise (Fig. 1(b))**: (window size=3, stride = 3) Map has disjoint block clusters . Equidistant patches aggregates local features. If we have two series $X_1$ and $X_2$ with length $N$, every 3 point construct one token, and the attention map is an $\frac{N}{3} \times \frac{N}{3}$  matrix.
> - **Window-wise (Fig. 1(c))**:  (window size=3, stride=1). Window-wise alignment provide more flexible split. If we If we have two series $X_1$ and $X_2$ with length $N$, every 1 point construct one token, and the attention map is an $N \times N$ matrix. But, we conduct the convolution with kernel size 3 on the attention map, so every grid of the attention map will aggregate its local 9 grids features.
>
> These differences lead to following attributes:
>
> - **Point-wise**:  This kind of method could not provide structure patterns for attention, which increase the burden on attention, making learning more difficult.
> - **Patch-wise**: For cross-domain generation, selecting one fixed window to satisfy all periodicity could be hard, and the unsuitable window will break some domain's periodicity.
> - **Window-wise**:  For every domain, there are expected to have a partial of windows can fit the specific periodicity for this domain. This could simplified the structure capturing process for attention mechanism.
>
> ---
>
> **Q2: Sensitive to kernel initialization & how it changes**
>
> > Q2: Are results sensitive to the convolution kernel initialization method? How do the kernel weights change along the training process?
>
> We have explored the changes in the kernel. Specifically, we tested two kernel initialization methods. One using random initialization, and the other using two-dimensional DCT bases for initialization.
>
> 1. **Sensitivity to the convolution kernel initialization**:The visualizations for kernel with random initialization are shown in Fig.6(a) (added in Section 5) and Fig.6(b) (added in Section 5), which are the kernel weights before and after training.  It can be observed that the values in the weight matrix tend to be evenly distributed. This means the convolutional layer reduced during the learning process, and it gradually transformed as ‘kernel’ of the average pooling layer, whose weights for each position are equal. Thus, the empirical results of random initialized kernel **tends to estimate the results of average pooling** after training. Beyond visualization analysis, we also provide numerical results shown in the following table. From the table, we can find that the results of random initialized kernel close to that of the average pooling, with decrease in 5 datasets and increase in 7 datasets.
>
> **Table 1.1 The comparative results with average pooling and random initialized convolution method.** The empirical results of random initialized kernel tends to estimate the results of average pooling after training
>
> |             | avg_pool | random_conv |
> | ----------- | -------- | ----------- |
> | Electricity | 0.002    | **0.001**       |
> | Solar       | **0.033**    | 0.039       |
> | Wind        | 0.037    | **0.025**       |
> | Traffic     | 0.074    | **0.071**       |
> | Taxi        | 0.082    | **0.079**       |
> | Pedestrain  | 0.040    | **0.035**       |
> | Air         | **0.012**    | 0.014       |
> | Temperature | **0.229**    | 0.237       |
> | Rain        | **0.036**    | 0.048       |
> | NN5         | 0.154    | **0.143**       |
> | Fred-MD     | **0.002**    | 0.004       |
> | Exchange    | 0.138    | **0.137**       |
> | Best Count  | 5        | 7           |
>
> 2. **How do the kernel weights change?**: The visualizations for kernel initialized with DCT are shown in Fig.6 (c) (added in Section 5) and Fig.6(d) (added in Section 5). It can be observed that the weights have **minor changes after training**, because the DCT basis already has a good effect on capturing periodicity.  We can infer that DCT basis is a good tools to capture the periodicity, and the weights is already beneficial for attention map. Thus, it changes slightly.

---

> ### Author Response · Authors · 2025-11-20
> **Official Comment by Authors**
>
> **Q3: Hyper-parameter experiments & Rules for hyper-parameter selection**
>
> > Q3: As the results for different kernel sizes are close to each other, can the authors provide some general rules for selecting this hyper-parameter?
>
> We report the hyper-parameter about the window size in Appendix A. According to the following table, we **summary some rules** for kernel selection.
>
> Step 1: Choose Kernel Type: We can find that **DCT kernel performs better than Avg pooling** (DCT achieves **8+8+9=25** best counts, while Avg achieves **4+9+5=18** best counts), because DCT kernel could capture more periodicity information. Thus, in this paper, we mainly consider the DCT kernels. By the way, **Avg Pooling is more efficient**, so it's more suitable with limited calculation consumption.
>
> Step 2: Choose Kernel Size: The **kernel size with 25 for DCT** is recommended, as it could **cover common cycles** of these datasets. The optimal 25-size kernel in the paper works because it includes typical cycles (4, 6, 12, 24) without overcomplicating computation. Avoid sizes smaller than the smallest intrinsic cycle of the target dataset, e.g., <4,  to prevent truncating fine-grained periodicity. Larger sizes are preferred but not too large, which may compress features excessively leads to the loss of time scales.
>
> **Table 1.2 Hyper-parameter studies.**
>
> | MMD         | conv,ks=7 | conv,ks=13 | conv,ks=25 | avg.ks=7 | avg,ks=13 | avg,ks=25 |
> | ----------- | --------- | ---------- | ---------- | -------- | --------- | --------- |
> | Electricity | **0.001**     | 0.002      | **0.001**      | 0.002    | 0.002     | 0.002     |
> | Solar       | 0.035     | 0.035      | 0.035      | **0.033**    | **0.033**     | **0.033**     |
> | Wind        | 0.035     | **0.033**      | 0.034      | **0.033**    | **0.033**     | 0.037     |
> | Traffic     | 0.074     | 0.074      | **0.071**      | 0.072    | **0.071**     | 0.074     |
> | Taxi        | **0.080**     | 0.086      | 0.085      | 0.084    | 0.083     | 0.082     |
> | Pedestrain  | **0.040**     | 0.041      | **0.040**      | 0.042    | 0.041     | **0.040**     |
> | Air         | 0.014     | 0.013      | **0.011**      | 0.013    | 0.013     | 0.012     |
> | Temperature | 0.226     | **0.219**      | 0.230      | 0.220    | **0.219**     | 0.229     |
> | Rain        | **0.033**     | 0.050      | 0.036      | 0.038    | 0.037     | 0.036     |
> | NN5         | 0.149     | 0.151      | **0.147**      | 0.153    | 0.154     | 0.154     |
> | Fred-MD     | **0.002**     | **0.002**      | **0.002**      | **0.002**    | **0.002**     | **0.002**     |
> | Exchange    | **0.136**     | 0.139      | 0.137      | 0.140    | 0.139     | 0.138     |
> | KL          | conv,ks=7 | conv,ks=13 | conv,ks=25 | avg.ks=7 | avg,ks=13 | avg,ks=25 |
> | Electricity | 0.019     | 0.020      | **0.008**      | 0.033    | 0.023     | 0.014     |
> | Solar       | 0.021     | **0.012**      | 0.013      | 0.015    | 0.015     | 0.016     |
> | Wind        | 0.201     | 0.186      | 0.202      | **0.180**    | 0.182     | 0.203     |
> | Traffic     | 0.013     | **0.009**      | 0.011      | 0.010    | **0.009**     | 0.011     |
> | Taxi        | 0.006     | **0.004**      | 0.005      | 0.005    | **0.004**     | **0.004**     |
> | Pedestrain  | **0.009**     | 0.011      | **0.009**      | 0.012    | 0.012     | 0.010     |
> | Air         | 0.031     | 0.025      | 0.026      | 0.028    | 0.027     | **0.023**     |
> | Temperature | **0.173**     | 0.174      | 0.176      | 0.178    | 0.181     | 0.186     |
> | Rain        | 0.014     | **0.009**      | 0.011      | 0.011    | 0.015     | 0.012     |
> | NN5         | 0.048     | 0.049      | **0.045**      | 0.047    | 0.054     | 0.053     |
> | Fred-MD     | 0.248     | **0.197**      | 0.201      | 0.201    | **0.197**     | 0.204     |
> | Exchange    | 1.629     | 1.594      | 1.621      | 1.610    | **1.590**     | 1.635     |
> | BestCount   | 8         | 8          | 9          | 4        | 9         | 5         |

---

### Author Response · Authors · 2025-11-30
**Summary for the Responses**

We sincerely thank all reviewers for their insightful comments and constructive feedback, which have helped refine our work.

We are encouraged by the positive feedbacks, including:

- Our work focuses on a **meaningful and challenging task** relevant to broader applications. (Reviewer gSxW, tGxi)
- We propose a **novel and interesting method**, mitigating the limitation from traditional pair-wise or patch-wise methods. (Reviewer  gSxW, tGxi, 6vwQ)
- Our work shows **theoretical insights**, which provides groundings for the proposed Ample attention. (Reviewer  gSxW, tGxi)
- Extensive **experiments show impressive results**, demonstrating the method’s robustness and broad applicability, with discussions clarifying its advantages. (Reviewer  gSxW, 6vwQ, z9c8)
- The paper is **well-structured**, clear presented and easy-to-follow. (Reviewer z9c8)

The reviewers' comments have been helpful and we have responded these questions point-by-point in the following respects:

- **Structure and syntactic refinement:** Refined the sentences and figures for clarity, as suggested by gSxW, tGxi.

- **Motivation**: Clarified the motivation and theoretical groundings, as suggested by gSxW, tGxi.

- **More Visualization**: Appended the visualization and relevant discussion to response tGxi, gSxW, including:

  - Visualization for the changes of the kernel weights (Fig 6, Sec 5),  to explore how the kernel changes, which could verify the theoretical findings.
  - Visualization of generated samples among methods for comparison among methods (Fig 9, App C.2), which could verify the motivation.

- **Results**: Provided new experimental results and relevant discussion to response gSxW, tGxi, 6vwQ and z9c8, including:

  - More **sensitivity studies** on kernels to explore how these settings affect results (Table 1.1, Table 2.3/Table 4.1).

  - Integrating our Ample attention into forecasting pipelines, to verify the **scaliblity** (Table 2.1).

  - Evaluation through **downstream tasks** (Table 2.4/Table 3.1/Table 4.2) and **domain discrepancy** (Fig 12, App D) measurement, for more more **diversified evaluation**.

    *Other experimental results are selected from the original version of the appendix (Table 1.2, Table 2.2).*

---

### Meta-Review · Area_Chair_xfDc · 2026-01-03

**Summary:**

This paper proposes Winformer, a diffusion model framework designed for cross-domain time-series generation. Its core innovation lies in the introduction of a window-based attention mechanism (Ample Attention), which extends attention computation from point-to-point similarity to comparisons between continuous windows to better capture cross-domain periodic patterns. Extensive experiments conducted on 12 real-world datasets demonstrate that Winformer outperforms existing methods by an average of 10.67%. Reviewers generally agreed that the method is innovative, theoretically well-supported, and experimentally comprehensive. However, some deficiencies were noted regarding writing clarity, motivational explanations, and experimental scope (e.g., a lack of downstream task evaluation). In the rebuttal, the authors provided point-by-point responses, incorporating visualizations, theoretical derivations, hyperparameter studies, and downstream task experiments. Despite these efforts, two limitations persist: First, regarding the lack of novelty raised by Reviewer tGxi, the authors' additional theoretical derivations and motivational explanations fail to refute the concern that Ample Attention is built upon established ideas and that the overall architecture echoes recent works. Second, regarding the lack of downstream utility experiments raised by Reviewer 6vwQ, the authors' decision to test prediction tasks only on subsets of datasets with strong periodicity is insufficient to fully demonstrate the practical utility of the design across diverse downstream tasks.

**Reviewer Concerns:**

1) Reviewer gSxW (Initial Score: 4)
Issues Resolved: Fully resolved.

Score Change After Rebuttal: Expected increase.

Core Contributions Recognized: Acknowledged the significance of the method for cross-domain time-series generation; considered the window attention mechanism well-designed, the experiments comprehensive, and the theoretical derivations explanatory.

Key Responses and Resolutions: The reviewer raised concerns about incoherent writing, the lack of a conclusion, insufficient motivation, and a lack of comparative visual analysis. The authors addressed these by revising the language throughout the manuscript, adding a conclusion section (Section 7), providing multiple new comparative visualizations (Figs. 6, 8, 9), and supplementing theoretical motivation (Fig. 13). Computational overhead was also clarified in Appendix B. The response was systematic and substantial, fully addressing the reviewer's written critiques.

2) Reviewer tGxi (Initial Score: 4)
Issues Resolved: Not fully resolved.

Score Change After Rebuttal: Expected to remain unchanged.

Core Contributions Recognized: Recognized the theoretical value of the research problem and the window attention mechanism; affirmed that experiments demonstrated performance improvements.

Key Responses and Resolutions: In response to concerns regarding narrow applicability, lack of downstream task evaluation, and hyperparameter sensitivity, the authors added: 1) Experiments applying Ample Attention to prediction tasks (Table 2.1) to show potential in non-generative tasks; 2) Evaluations of prediction tasks using data augmentation with generated data (Table 2.4, etc.); 3) A detailed hyperparameter sensitivity analysis (window size, stride). These new analyses directly address the reviewer's core concerns. However, regarding the lack of novelty, while the authors provided theoretical derivations and motivational explanations, the concern remains that the mechanism is built on established ideas and the architecture closely mirrors recent work.

3) Reviewer 6vwQ (Initial Score: 6)
Issues Resolved: Partially resolved.

Score Change After Rebuttal: Expected to remain unchanged.

Core Contributions Recognized: Recognized the innovation, technical soundness, and robustness of the experiments.

Key Responses and Resolutions: The reviewer’s primary concern—the lack of proof regarding downstream utility—was addressed through new data augmentation prediction experiments (Table 3.1). These results show that augmenting with Winformer-generated data improves prediction model performance across several datasets. However, testing only on datasets with strong periodicity does not provide sufficient evidence for the design's utility across a broader range of downstream tasks, leaving the reviewer's concern partially unaddressed.

4) Reviewer z9c8 (Initial Score: 6)
Issues Resolved: Fully resolved.

Score Change After Rebuttal: Expected to remain unchanged.

Core Contributions Recognized: Affirmed the clear structure, outstanding experimental results, and overall research value.

Key Responses and Resolutions: All three questions raised were addressed specifically: 1) A comparative analysis of point/patch/window operations was added; 2) The choice of evaluation metrics (MMD, KL) was justified, and domain difference assessments were supplemented (Fig. 12); 3) A comparative analysis with TimeDP was provided, discussing the trade-offs between global structure capture and high-frequency noise. The reviewer commented after the rebuttal, stating they "maintain a positive score," indicating general satisfaction.

**Reviewer Scores:**

1) Reviewer gSxW (Initial: 4) - Potential Final Score: Likely 6 - Reason: The authors systematically addressed all three categories of concerns (textual/structural issues, motivation/theory, and comparative analysis/visualization). By providing new visualizations, deepening theoretical derivations, and adding hyperparameter experiments, the clarity and persuasiveness of the paper have significantly improved.

2) Reviewer tGxi (Initial: 4) - Potential Final Score: Likely 4 - Reason: Although the authors provided detailed responses regarding scope, novelty, and validation—including prediction task results and domain difference metrics—the fundamental concern regarding incremental innovation remains. The rebuttal failed to prove that the architecture is sufficiently distinct from established ideas and recent works.

3) Reviewer 6vwQ (Initial: 6) - Potential Final Score: Maintain 6 - Reason: The reviewer focused on downstream utility. While the authors showed an average MAE improvement of 29.62% in data augmentation experiments, the limited selection of datasets (strong periodicity only) prevents a complete confirmation of the model's general utility across diverse downstream tasks.

4) Reviewer z9c8 (Initial: 6) - Potential Final Score: Maintain 6 - Reason: The authors provided point-by-point responses to concerns about operational differences, metric selection, and comparisons with TimeDP. The reviewer explicitly stated their intention to maintain a positive score following the rebuttal.

---

### Decision · Program_Chairs · 2026-01-26

Reject